# Organo Mineral Fertilizers Increases Vegetative Growth and Yield and Quality Parameters of Pomegranate cv. Wonderful Fruits

Annalisa Tarantino [1,*], Grazia Disciglio [1], Laura Frabboni [1] and Giuseppe Lopriore [2]

[1] Department of Agriculture, Food, Natural Resources and Engineering (DAFNE), University of Foggia, Via Napoli 25, 71122 Foggia, Italy

[2] Department of Soil, Plant and Food Sciences, University of Bari Aldo Moro, Via G. Amendola 165/A, 70126 Bari, Italy

* Correspondence: annalisa.tarantino@unifg.it

**Abstract:** In recent years, to improve sustainable production in horticultural crops, many new types of strategies have been developed, including organo-mineral fertilization to complement chemical fertilizers in order to enhance the nutritional status of plants and sustainability of the agroecosystems. This study was performed on a young pomegranate orchard of the "Wonderful" cultivar during the 2018 and 2019 seasons. The purpose was to evaluate the effects of three foliar applications (at the red ball, fruit setting, and fruit development stages) of four commercial organo-mineral fertilizers (Hendophyt®, Ergostim XL®, Siapton® 10L, and Allibio Rad®) on vegetative growth, yield, and several other physico-chemical parameters of the fruits, throughout each season. The results obtained showed several differences between the two years. The annual trunk growth of trees under all compared treatments showed significantly lower values in 2018 (average 9.7 mm) than in 2019 (average 11.8 mm). At the end of the two-year period, the biostimulant treatments resulted in significantly larger trunk diameters (average 43.6 mm) than the control (39.6 mm). Only in 2018, significantly higher number of fruits per tree, number of arils per fruit, edible part, and juice yield were obtained under biostimulant treatments compared with the control. No differences among treatments were observed for any color parameters or physico-chemical traits in the fruits for either year. In 2019, fruit morpho-pomological properties tended to be lower than in 2018, while in contrast, total phenol content and antioxidant activity were higher. The warmer and windier weather conditions of 2019 probably led to greater plant stress conditions, with a reduction in fruit size and an increase in the bioactive compounds of juice. In conclusion, due to the various positive results, foliar organo-mineral fertilizers could be recommended to improve the performance of pomegranate Wonderful cv. under similar conditions.

**Keywords:** pomegranate; foliar spray; humic and fulvic acids; carboxylic acids; amino acids and peptides; polyglucosamine; morpho-physico-chemical properties



## 1. Introduction

The cultivation of pomegranate (*Punica granatum* L.) worldwide has increased sharply in the past few years, mainly due to the growing perception that this fruit has numerous medical benefits [1]. In recent years, in Italy, the pomegranate crop has also been successful; 1240 ha are currently cultivated, with a total production of 19,248 tons [1]. Indeed, although production continues to increase, it is still insufficient to meet internal needs. Apulia is one of the Italian regions most affected by this sudden expansion, with the current production of 33.8% of the national total [2], and the most cultivated variety being the "Wonderful" [3].

As is the case for other perennial fruit-tree species, pomegranate trees also require fertilization as an important management tool for increasing growth and quanti–qualitative yield parameters. It is, therefore, equally important to prevent any nutritional stresses other

than those caused by abiotic factors (e.g., drought, soil salinity, and different climatic parameters) to which this crop can be exposed [4–6]. Pomegranate cv. Wonderful encounters a number of serious problems such as fruit cracking, sunburn, lack of appropriate peel and pulp color, and the lowest yield average [7]. Plant production should be based on stimulating plant growth and development, while simultaneously reducing risks posed to humans and the natural environment, as well as providing safe, high-quality agricultural products [8]. To deal with these problems, conventional agriculture must become less dependent on chemical fertilizers, which have a serious impact on the natural ecosystem and human health. There are several kinds of nutrients necessary for plant growth; however, nitrogen (N) and phosphorus (P) are two of the most important and abundant. For instance, the pomegranate fruit crop requirement is approximately 0.25 Kg of N per 100 Kg of fruit harvested [9]. N and/or P are critical determinants of plant growth in most ecosystems. Synthetic fertilizers can harm the environment because their N and P levels are often higher than those in natural soils. Excessive use of nitrogen can impact not only the climate, but can also damage plant health and reduce biodiversity, both on land and in our waterways [10]. The long-term land application of phosphorous-enriched fertilizers leads to phosphorus accumulation in soil that may become susceptible to mobilization via erosion, surface runoff, and subsurface leaching [11]. In this regard, the aim of modern agriculture is to use new sustainable technology dedicated to the sustainable development agroecosystems [12]. Fertilization techniques have also, therefore, moved towards organic, sustainable, and environmentally friendly strategies, to complement chemical fertilizers in order to improve plant nutrition [13,14]. In this context, "Agricultural Biostimulants" (ABs) are used to support crop growth and yield, and to improve the final quality of products [15–17]. ABs are mostly natural fertilizer defined as "any substance or microorganism applied to plants with the aim to encourage growth, enhance nutrition efficiency, abiotic stress tolerance and/or crop quality traits, regardless of its nutrients content" [18]. The ABs include natural substances such as humic and fulvic acids, protein hydrolysates, nitrogen-containing compounds, seaweed extracts, beneficial fungi and beneficial bacteria, chitosan and other biopolymers, and inorganic compounds [17–23]. These can generally be used in both conventional, integrated, and organic agriculture [24], and are applied to the soil and/or by foliar spraying. In the latter case, ABs contain small soluble molecules [19] which, if implemented in specific growth periods, represent a refined technique to optimize plant health and production quality [25]. Numerous studies showed the positive effect of ABs on the growth, yield and/or fruit quality of peach and apple [26,27], apricot [28–30], mandarin [31], kiwifruit [32], table grape [33–35], olive [36], strawberry [37] or enhanced root development in grape [38,39], pear trees [40], and strawberry [41,42]. The effect of foliar-applied biostimulant substances on the yield and fruit quality of pomegranates has been studied in several cultivars, but information about this effect on pomegranate cv. Wonderful is very scarce. Some research has been undertaken into the applications of chemical and organic mixtures, amino acids, or biofertilizers on certain cultivars (e.g., 'Manfalouty', 'Kandhari', 'Ambrosia') to limit or prevent fruit cracking [22,43–47], to increase fruit retention and yield [48], to improve the growth and yield of trees, and to improve the biochemical quality of the fruit [49–52]. Moreover, microbial biofertilizers improve plant height, plant canopy, pruned materials, and fruit yield in pomegranate [53].

Research on this cultivar reported that foliar applications, including ascorbic acid with amino acids, vitamin B, and silicon improve the growth, yield, and quality of pomegranate trees [54,55], or improve fruit mineral nutrient concentrations, bioactivity, and internal quality [1,56]. As for the nutritional and chemical content of the pomegranate fruit, various data are reported in our previous experimental studies [57–59].

The aim of this study was to evaluate in the semiarid environment of the Apulia region of Italy, also characterized by high wind speeds, the effects of foliar application of four commercial organo-mineral fertilizers (Hendophyt® PS, Ergostim® XL, Siapton10L® and AlliBio-Rad®) on vegetative growth, yield, and some physicochemical parameters of pomegranate Wonderful cv., in order to reduce fertilizer application rates and enhance

plant nutrient-use efficiency. Furthermore, the application of organo-mineral fertilizers as biostimulants in horticultural crops can be a particularly valuable tool in agroecological and sustainable agriculture.

## 2. Materials and Methods

### 2.1. Location and Biostimulant Treatments

Field experiments were conducted over two consecutive seasons (2018 and 2019) in a young commercial orchard of pomegranate cv. "Wonderful" (which corresponded to 3- and 4-year-old plants) located in the Foggia countryside (Apulia region, southern Italy, 41°27′08″ N, 15°31′56″ E) and at an altitude of 54 m a.s.l. The soil is a silty-clay vertisol of alluvial origin (1.20 m depth) (Typic Chromoxerert, fine, thermic, according to the Soil Taxonomy-USDA), with the following characteristics: 32.7% clay; 30.5% silt; 36.8% sand; pH 8.8 (soil:water 1:2.5); 1.4% organic matter; 1.5‰ total N; 9.7% active $CaCO_3$; 56 mg kg$^{-1}$ $P_2O_5$; and 1390 mg kg$^{-1}$ exchangeable K.

For each season, the four different commercial ABs, Hendophyt® PS (Iko-Hydro), Ergostim® XL (Isagro), Siapton10L® (Siapa), and AlliBio-Rad® (Fertek) with biostimulant action were compared to a control (sprayed with water). Table 1 presents the composition, including the main active compounds and the dose of different products used in the trials. In particular, these contain polysaccharide biopolymers (polyglucosamine), carboxylic acids (N-acetylthiazolidin-4-carboxylic acid -AATC and triazolidinecarboxylic acid -ATC), humic and fulvic acids and amino acids (protein hydrolysates), which are among the various substances which cause a bio-stimulating action in plants [6]. The products were applied, in each year, by foliar spraying three times during the growing season (at the flower bud, fruit setting, and fruit growth stages) on 29 May, 29 June, and 13 September 2018, and 5 June, 8 July, and 4 September 2019. All treatments were performed with a total volume of 250 L ha$^{-1}$ and each tree was sprayed until the run-off point using a pulled sprayer under favorable weather forecasts (no rainfall expected in the following 24 h). The experiment set up was organized as a completely randomized block design with four replications per treatment and five trees per replicate. Trees were planted at a distance of 3 m on the row and 5.5 m between the rows.

**Table 1.** Formulations and doses of foliar applications of Agricultural Biostimulant (ABs) commercial products used in the experiment.

| ABsTreatments |
| --- |
| **HENDOPHYT PS** (Iko-Hydro): a fully water-soluble powder, comprising biopolymers of polysaccharides (polyglucosamine) 60%, containing carbon 35%, organic nitrogen 4%, boron 0.25%; applied at a dose of 150 g 100 L$^{-1}$ of water. |
| **ERGOSTIM XL** (Isagro): a concentrated water-soluble liquid _N-acetiltiazolidin-4-carboxylic acid (AATC) 2.5%, and triazolidine-carboxylic acid (ATC) 2%; i applied at a dose of 200 mL 100 L$^{-1}$ of water. |
| **SIAPTON 10L** (Siapa): based on amino acids and peptides originating from chemical hydrolysis of animal epithelium, with a high content of proline, hydroxychlorine, glycine and arginine; containing organic nitrogen 8.7%, carbon 25%; C/N ratio = 2.9%; applied at a dose of 300 mL 100 L$^{-1}$ of water. |
| **ALLIBIO-RAD** (Fertek): a suspension–solution of humic and fulvic acids, obtained from worm compost (night crawled). Dry composition: total organic matter 60%; extractable organic substance 6% of organic matter; humified organic substance 80% extractable organic matter; organic substance 1.5% of extractable organic nitrogen; C/N ratio = 20; pH 8; applied at a dose of 150 g 100 L$^{-1}$ of water. |

To avoid any contamination between treatments, one buffer row was located between replicates and blocks, along with two or more buffer rows around the perimeter of the experimental field. In each replicate, three centrally located plants per plot were used to collect the vegetative and reproductive parameters.

### 2.2. The Climate

The weather conditions between the two seasons were quite different (Table 2). In fact, although the minimum temperatures for both seasons were similar (average 15.9 °C in 2018 and 15.4 °C in 2019), the maximum 2019 temperatures were warmer (average temperature 28.8 °C) than those in 2018 (average temperature 23.7 °C). In particular, in 2019, the months of June, July, and August were warmer (average maximum temperature 26.5 °C) than in 2018 (average maximum temperature 24.6 °C). Furthermore, in 2019, the overall weather was windier and drier (wind speed 3.7 m s$^{-1}$, and total precipitation 175.7 mm, respectively) than in 2018 (wind speed 2.9 m s$^{-1}$, total precipitation 366.4 mm, respectively).

**Table 2.** Monthly mean maximum and minimum temperatures ($T_{max}$, $T_{min}$) and relative air humidity ($RH_{ma}$ and $RH_{min}$), wind speed ($W_s$), radiation (Rad) and total precipitation (P) for the 2018 and 2019 seasons.

| Month | $T_{max}$ (°C) | $T_{min}$ (°C) | $RH_{max}$ (%) | $RH_{min}$ (%) | $W_s$ (m s$^{-1}$) | Rad (Wm$^{-2}$) | P (mm) |
|---|---|---|---|---|---|---|---|
| **2018** | | | | | | | |
| April | 21.3 | 12.9 | 94.6 | 37.6 | 2.8 | 235.3 | 54.0 |
| May | 26.1 | 13.4 | 95.2 | 49.1 | 2.4 | 275.8 | 58.3 |
| June | 30.0 | 12.1 | 89.5 | 40.3 | 3.4 | 289.6 | 88.2 |
| July | 33.3 | 19.6 | 83.6 | 35.4 | 3.0 | 318.7 | 16.8 |
| Aug | 32.7 | 20.1 | 71.3 | 28.3 | 2.1 | 285.7 | 39.1 |
| Sept | 29.1 | 17.1 | 81.3 | 30.0 | 3.7 | 193.6 | 80.0 |
| **Mean** | **23.7** | **15.9** | **85.9** | **36.8** | **2.9** | **266.5** | |
| **Total** | | | | | | | **366.4** |
| **2019** | | | | | | | |
| April | 20.6 | 8.2 | 94.4 | 51.0 | 3.7 | 190.2 | 40.3 |
| May | 21.3 | 10.2 | 95.3 | 56.3 | 4.0 | 232.9 | 86.7 |
| June | 33.2 | 17.5 | 85.9 | 35.1 | 3.7 | 252.2 | 9.2 |
| July | 33.7 | 19.5 | 84.0 | 33.9 | 3.7 | 258.8 | 30.0 |
| Aug | 34.8 | 20.3 | 79.9 | 33.9 | 3.6 | 225.6 | 5.7 |
| Sept | 29.5 | 16.8 | 88.7 | 42.6 | 3.6 | 175.5 | 3.8 |
| **Mean** | **28.8** | **15.4** | **88.0** | **42.5** | **3.7** | **222.5** | |
| **Total** | | | | | | | **175.7** |

### 2.3. Vegetative Growth Parameters

The trunk diameter (TD) and shoot length (SL) were determined for both seasons on three plants in each treated plot. The TD was measured four times in 2018 (8 April, 28 May, 27 June, and 17 September) and three times in 2019 (5 June, 12 September, and 28 October). It was measured at a marked point, 50 cm above the ground level, using a Vernier digital caliper, and expressed in millimeters. The annual trunk growth (ATG) and the trunk cumulative growth (TCG) were also calculated. The ATG was determined by calulating the difference in the measurement made in October and that in April each year; whereas TCG was determined by calculating the difference between the measurement made on 25 October 2019 and that on 5 April 2018.

The SL for each year was determined at bud break (in April) and at the end of the growing season (in October) on four tagged one-year-old shoots in the four-cardinal direction (East, West, North, and South) using a tape, and expressed in centimeters.

### 2.4. Yield, Morpho-Pomologicaland Physico-Chemical Parameters, Polyphenols Content

Pomegranate fruits were harvested on a single date (on 18 October 2018 and on 29 October 2019). At harvest time, fruits per tree for each plot were counted and weighed. The average number and weight (kg) of fruit per tree and yield (ton ha$^{-1}$) were calculated. To determine fruit quality characteristics, 12 sample fruits from each treatment were randomly

collected and stored in a portable ice box to be carried to the laboratory and maintained at 5 °C in a cold storage room for no longer than 72 h prior to analysis.

The following morpho-pomological traits were measured e/o calculated on the fruits: weight (g), length (mm), equatorial diameter (mm), percentage of fruit with a diameter of less than 80 mm (%), and color index of the epicarp (CIE coordinates, L*, a*, and b*) along the equatorial axis at two opposite spots. Following this, four fruits from each treatment, randomly chosen, were cut longitudinally and arils for each fruit were manually separated from non-edible fractions (skin, membrane and albedo). The following characteristics were then analyzed: skin weight (g), number (n.) and weight (g) of arils per fruit, aril edible portion (%), length (mm) and width (mm) of arils, fresh weight of 100 arils (g) and aril dry weight (%).

In addition, the following physical, chemical, and phytochemical analyses of juice were also performed: juice yield ($cm^3$ 100 $g^{-1}$ of arils), sugars content (TSS, °Brix), pH, total acidity (TA % of citric acid) and maturity index (MI) as the ratio °Brix/total acidity, total phenols (mg GA 100 $g^{-1}$ fresh weight), and antioxidant capacity (mg TE 100 $g^{-1}$ fw). All the above parameters were evaluated using the methods and instruments as previously fully described for pomegranate [57–59].

### 2.5. Statistical Analysis

Morpho-pomological and colorimetric analyses of fruit were performed on twelve samples, and physical, chemical, and polyphenol analyses of juice were performed on four samples. For each sample, three analytical replicates were made. The results were evaluated with a one-way ANOVA using the JMP® software package, version 14.3 (SAS Institute Inc., Cary, NC, USA) and average values were compared in a Tukey test. Standard deviations (SD) were calculated using Excel® software from the Office 2007® suite (Microsoft Corporation, Redmond, WA, USA). Percentage values were transformed to arcsine prior to analysis of variance.

## 3. Results and Discussion

### 3.1. Vegetative Growth Evaluation

The annual trunk growth (Figure 1) for all compared treatments shows significantly lower values in 2018 than in 2019. Differences were found among the treatments being compared. In particular, in the first year, significantly larger diameters were found in the Ergostim XL®, Siapton 10L®, and Allibio-Rad® treatments (11.0, 10.8, and 10.0 mm, respectively) compared with both the Hendophit Ps® and control (8.8 and 8.0 mm, respectively). In the second year, there was no significant difference among the biostimulant treatments (with values ranging from 11.9 to 12.5 mm); however, the treatments were all statistically higher than the control (10.0 mm).

Trunk cumulative growth (TCG) (Figure 2) gradually increased with some differences among treatments and the two seasons. After the two consecutive seasons, the largest TCG (44.9 mm) was noted for the Siapton 10L® treatment, not significantly different to the Allibio-Rad® (43.9 mm), Ergostim Xl® (43.6 mm), or Hendophit Ps®, but statistically different to the control (39.6 mm). These results show the sensitivity of pomegranate tree stem growth to biostimulant application. Indeed, the use of foliar fertilizers containing macromolecular compounds, in addition to the typical chemical elements (carbon, oxygen, hydrogen, nitrogen, sulfur), positively influenced tree growth as these enhance metabolism and plant photosynthesis activity [60].

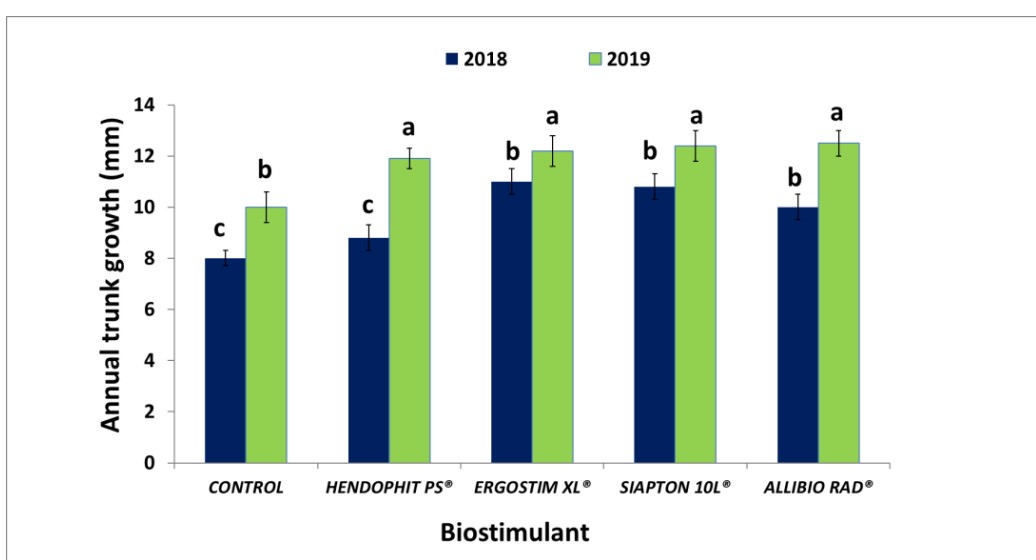

**Figure 1.** Annual trunk growth under biostimulant and control treatments in the 2018 and 2019 growing seasons. Average values ± SD with different letters between seasons and treatments indicate significant differences according to Tukey's test (*p* < 0.05).

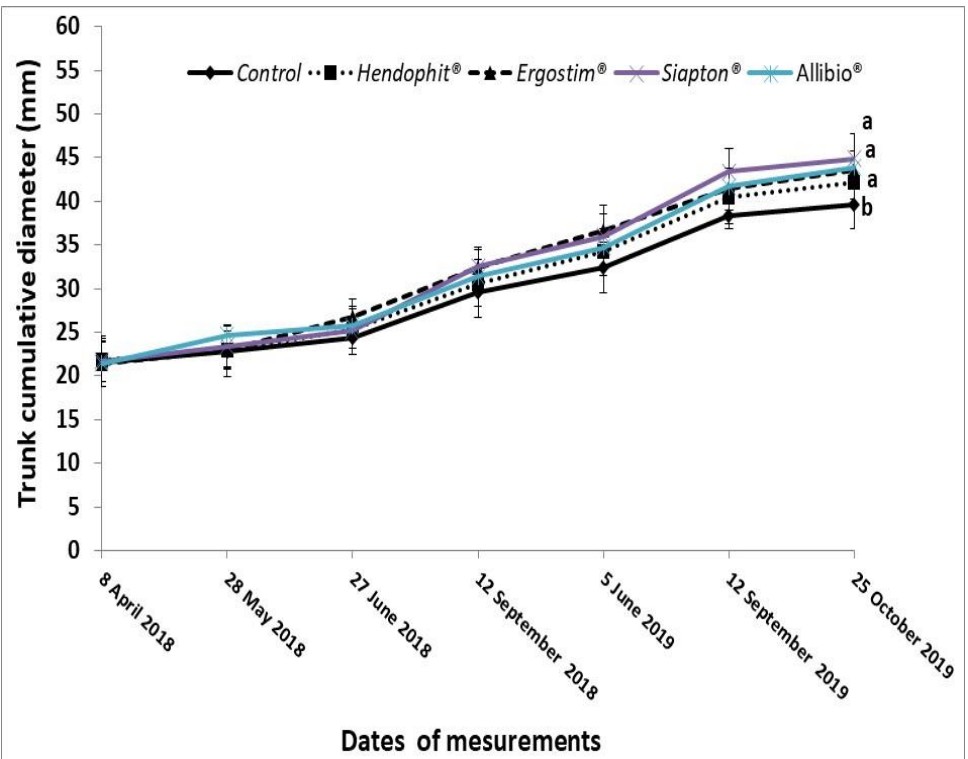

**Figure 2.** Trunk cumulative growth for each biostimulant and control treatment during the two seasons from April 2018 to October 2019. Treatments with different letters at the end of the two seasons were significantly different at *p* ≤ 0.05, according to the Tukey's test.

As for shoot development, no significant differences were noted among the four cardinal directions of the trees or for the different treatments (data not shown). However, the average shoot length tended to be higher under the biostimulant treatments (17.6 cm in 2018 and 37.1 cm in 2019) in respect to the control (14.2 cm in 2018 and 26.2 cm in 2019) (Figure 3). These results are in accordance with a previous study [55], where the foliar application of humic and fulvic acids provided greater elongation of the shoots.

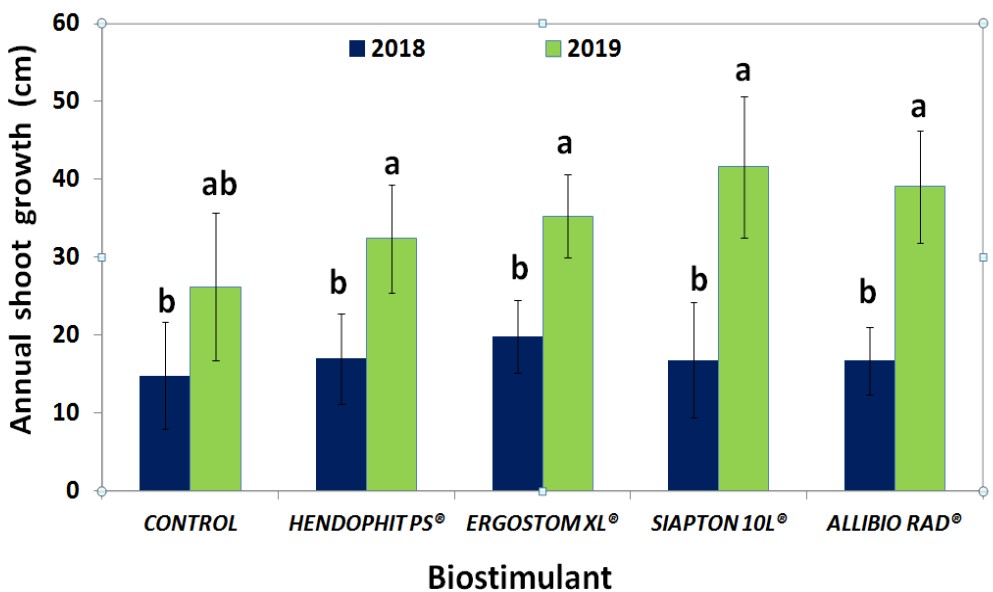

**Figure 3.** Annual shoot growth. Average values ± std. dev. in 2018 and 2019 are shown. Different letters per year indicate significant differences at $p < 0.05$, according to the Tukey's test.

*3.2. Yield Evaluation*

The average numbers of fruits per tree ($n_o$) and yield per tree (kg), according to the total harvest, were basically higher in 2019 than in 2018 because of the age of the plants (Figure 4). In 2018, the average number of fruits per tree showed non-significant differences among treatments (average = 16.2), while in 2019, the highest values were recorded when Wonderful pomegranate trees were treated with Hendophit Ps® (41.0), Ergostim Xl® (37.7), and Allibio-Rad® (33.1), the values of which were significantly higher than those obtained from both the Siapton 10L® (27.1) and control (23.2) treatments. Although they differed, the average yield per tree, in both years, was not significantly different among treatments; however, the biostimulants tended to show higher average values (5.2 kg/tree in 2018 and 9.9 kg/tree in 2019) than the control (3.9 kg/tree in 2018, and 7.9 kg/tree in 2019).

The difference in weather conditions over the years has influenced various production parameters and fruit quality. In particular, in the second year of the trial, characterized by higher maximum temperatures, greater windiness and drier conditions, the size of the fruit and the individual constituents of the fruit itself tended to be smaller; in contrast, however, the physical–chemical parameters were higher than in 2018. These results were also previously confirmed by other studies on pomegranate trees, when extreme hot and dry conditions during the period of maximal growth were associated with restricted growth and a high proportion of small-size fruit [55].

*3.3. Morpho-Pomological Evaluation of Whole Fruits*

Although the data are not statistically different from each other, the average weight, diameter, and length of fruits in general tended to be lower in 2019 than in 2018 (Table 3). The warmest and windiest weather in the second year led to higher stress condition in the plants, resulting in a reduction in fruit size. These results are in accordance with a previous study [55]. In each season, the aforementioned parameters were not significantly different among treatments, however, the biostimulant applications tended to increase the average values (454.8 g, 89.2 and 79.9 mm, respectively) relative to the control (412.7 g, 86.0 and 74.5 mm, respectively).

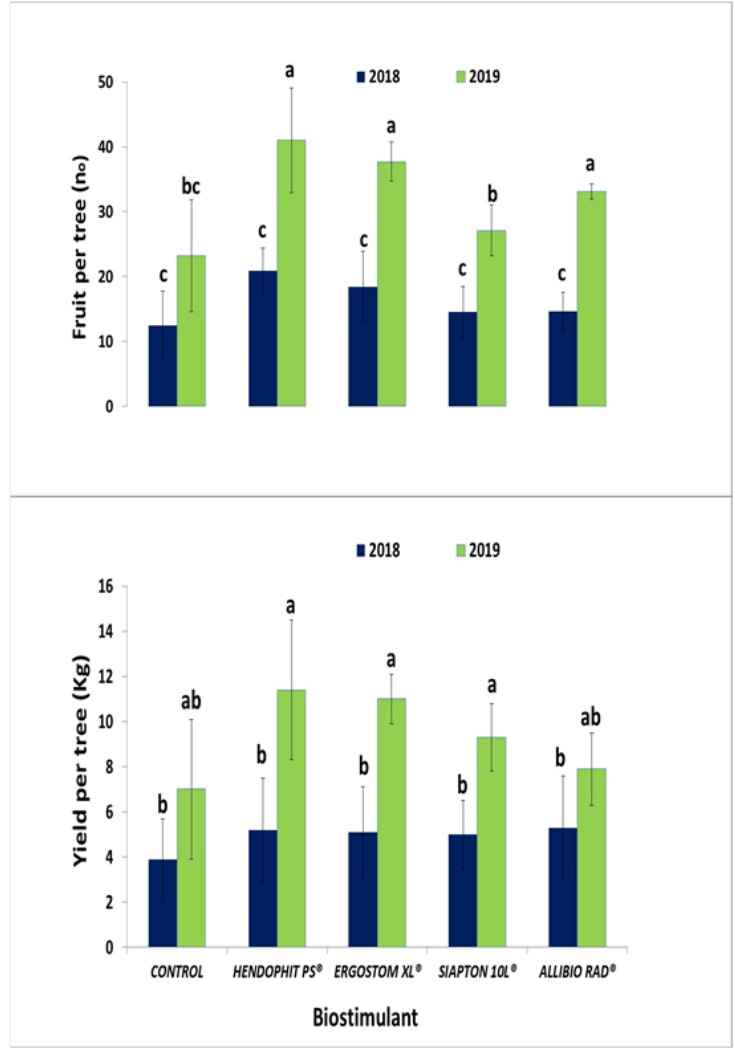

**Figure 4.** Average number of fruits per tree and yield per tree. Average values ± std. dev. in 2018 and 2019 are shown. Different letters per year indicate significant differences at *p* < 0.05, according to the Tukey's test.

**Table 3.** Effect of different biostimulants on morpho-pomological attributes of "Wonderful" pomegranate whole fruits.

| Parameter | Year | Biostimulant Treatment | | | | | Average Years |
|---|---|---|---|---|---|---|---|
| | | Control | Hendophit PS® | Ergostim XL® | Siapton 10L® | Allibio-Rad® | |
| Fruit average weight (g) | 2018 | 411.1 ± 72.8 | 457.0 ± 142.0 | 512.1 ± 106.2 | 476.0 ± 119.1 | 473.6 ± 124.6 | 465.9 ± 112.9 |
| | 2019 | 414.4 ± 87.3 | 409.3 ± 69.4 | 440.0 ± 74.6 | 477.1 ± 101.4 | 393.4 ± 59.8 | 426.8 ± 78. |
| Fruit diameter (mm) | 2018 | 86.5 ± 12.2 | 91.0 ± 10.6 | 89.5 ± 12.6 | 93.2 ± 9.3 | 86.8 ± 13.4 | 89.4 ± 11.6 |
| | 2019 | 85.6 ± 10.2 | 87.5 ± 8.2 | 90.8 ± 6.5 | 89.9 ± 6.6 | 84.8 ± 6.6 | 87.7 ± 7.6 |
| Fruit length (mm) | 2018 | 74.4 ± 10.6 | 81.9 ± 10.2 | 80.4 ± 9.1 | 82.9 ± 9.3 | 80.3 ± 9.1 | 79.9 ± 9.6 |
| | 2019 | 74.6 ± 12.2 | 76.6 ± 8.5 | 80.2 ± 6.8 | 81.9 ± 8.6 | 75.3 ± 7.2 | 77.7 ± 8.6 |

Average values ± std. dev. are not statistically different from each other.

### 3.4. Main Constituent Parts of Pomegranate Fruits

The data in Table 4 show that all parameters relating to arils, in agreement with the whole-fruit parameters, tended to be lower in 2019 than in 2018. Moreover, in both years, the aril weight per fruit and the fresh weight of 100 arils were not significantly different among all the compared treatments, the mean values of which ranged from 168.2 to 263.0 g and from 24.5 to 30.2 g, respectively. However, even these parameters show values that tended to be higher for the biostimulant treatments, than for the control.

**Table 4.** Effect of different biostimulants on the main constituent parts of pomegranate fruits of the cv. "Wonderful".

| Parameter | Year | Biostimulant Treatment | | | | | Average Years |
|---|---|---|---|---|---|---|---|
| | | Control | Hendophit PS® | Ergostim XL® | Siapton 10L® | Allibio-Rad® | |
| Aril weight per fruit (g) | 2018 | 210.8 ± 57.0 | 234.6 ± 81.4 | 263.0 ± 86.8 | 223.8 ± 71.5 | 230.9 ± 70.3 | 193.8 ± 73.4 |
| | 2019 | 184.2 ± 47.6 | 184.5± 61.3 | 207.7 ± 44.0 | 217.8 ± 63.0 | 168.2 ± 59.6 | 160.4 ± 55.1 |
| Fresh weight 100 arils (g) | 2018 | 28.6 ± 4.4 | 28.8 ± 5.4 | 30.2 ± 3.1 | 29.0 ± 3.7 | 29.7 ± 4.7 | 24.4 ± 4.3 |
| | 2019 | 24.5 ± 1.7 | 28.7 ± 4.7 | 29.8 ± 1.2 | 24.8 ± 1.8 | 24.9 ± 4.3 | 22.1 ± 2.7 |
| Aril per fruit ($N_o$) | 2018 | 734.5 ± 19.6 b | 814.6 ± 25.9 a | 870.9 ± 28.6 a | 771.7 ± 23.4 b | 864.8 ± 25.0 a | 676.1 ± 24.5 A |
| | 2019 | 721.4 ± 16.1 c | 691.2 ± 15.3 c | 688.1 ± 18.4 c | 761.1 ± 14.6 b | 740.2 ± 16.1 b | 600.3 ± 14.1 B |
| Fruit edible portion (%) | 2018 | 48.2 ± 1.0 b | 51.3 ± 1.4 a | 51.3 ± 1.8 a | 49.5 ± 1.4 ab | 48.9 ± 1.3 ab | 41.5 ± 1.38 A |
| | 2019 | 44.3 ± 4.0 b | 48.5 ± 7.4 ab | 46.6 ± 5.0 ab | 45.6 ± 3.9 ab | 46.9 ± 4.3 ab | 38.7 ± 4.92 B |
| Juice volume (cm³ 100 g⁻¹) | 2018 | 81.1 ± 2.3 cd | 100.8 ± 2.9 a | 96.2 ± 2.3 ab | 94.2 ± 2.3 b | 95.6 ± 2.2 ab | 78.0 ± 2.4 A |
| | 2019 | 77.7 ± 2.1 cd | 97.3 ± 2.0 ab | 98.1 ± 1.7 ab | 82.3 ± 2.1 c | 81.3 ± 1.9 cd | 72.8 ± 1.96 B |
| Dry matter arils (%) | 2018 | 20.1 ± 0.4 | 20.3 ± 0.5 | 20.4 ± 0.4 | 20.5 ± 0.5 | 20.3 ± 0.5 | 16.9 ± 0.46 |
| | 2019 | 23.3 ± 1.0 | 22.3 ± 0.8 | 21.1 ± 1.1 | 22.1 ± 0.9 | 21.4 ± 0.9 | 18.4 ± 0.94 |

Average values ± std. dev. of 2018 and 2019 years and the relative average of two years are showed. Different lowercase letters per year, and different capital letters between year averages, both indicate significant differences at $p < 0.05$, according to the Tukey's test.

It was further noted that, among the biostimulant treatments, Hendophit Ps® provided the highest results for both the aforementioned parameters. A similar response was also verified in 2018 for arils per fruit, which were highest with Ergostim XL® (870.9). This was not, however, significantly different from to either Allibio-Rad® (864.8) or Hendophit Ps® (814.6) treatments, although was higher than the treatment with Siapton 10L® (771.7), which was, in turn, significantly different to the control (734.5). In 2019, the best results were obtained for both Siapton 10L® (761.1) and Allibio-Rad® (740.2) treatments, compared with any of the others (values between 721.4 and 691.2).

A similar trend was noted for the edible portion of the fruit, which, in the first season, showed the highest values for both the Hendophit Ps® (51.3%) and Ergostim XL® treatments (51.3%); these were not statistically different to either Siapton 10L® (49.5%) or Allibio-Rad® (48.9%), but they were both different to the control (48.7%). In the second season, the application of each biostimulant product exhibited values that tended to be higher (ranging from 45.6 to 48.5%) than in the control (44.3%), but not statistically different among them. It is well known that a high edible portion of fruit is a positive result from the perspective of the processing plants because it involves greater product yield and less waste production for the same quantity of pomegranate processed. One of the most important parameters from the juice industry viewpoint is the juice yield, nominally the juice volume obtained from 100 g of arils. The values in 2018 were higher for the Hendophit Ps® treatment (100.8 cm³ 100 g⁻¹), although not significantly different to either Ergostim XL® (96.2 cm³ 100 g⁻¹) or Allibio-Rad® (95.6 cm³ 100 g⁻¹), which in turn were not statistically different to Siapton 10L® (94.2 cm³ 100 g⁻¹); they were, however, different to the control (81.1 cm³ 100 g⁻¹). Moreover, in 2019, high values were obtained for both the Ergostim XL® and Hendophit Ps® treatments (98.1 and 97.3 cm³ 100 g⁻¹, respectively), which were significantly different from the others (ranging from 77.7 to 81.3 cm³ 100 g⁻¹). Our juice-yield mean values for the Wonderful cultivar fruits were similar to those reported by Ferrara et al. [61,62], but higher than those reported by Passafiume et al. [63]. Finally, no significant differences among treatments were observed for the aril dry weight (average 20.3%).

### 3.5. Colorimetric Characteristics

As indicated by the color data (Table 5), the mean color parameters (L*, a* and b*) for all treatments were higher in the skin than in the arils and juice. This is in agreement with

previous studies performed on several cultivars [63–65]. Concerning each component of the fruit (skin, aril, and juice) no differences among treatments were observed for any of the color parameters.

**Table 5.** Effect of different biostimulants on skin, aril, and juice color characteristics of "Wonderful" pomegranate fruits.

| Parameter | | Year | Biostimulant Treatment | | | | | Average Years |
|---|---|---|---|---|---|---|---|---|
| | | | Control | Hendophit PS® | Ergostim XL® | Siapton 10L® | Allibio-Rad® | |
| Skin | L* | 2018 | 42.1 ± 6.0 | 42.3 ± 5.5 | 43.1 ± 6.7 | 42.3 ± 5.3 | 42.2 ± 6.2 | 42.4 ± 5.9 |
| | | 2019 | 40.0 ± 2.6 | 42.6 ± 5.6 | 40.0 ± 2.1 | 40.2 ± 3.7 | 40.5 ± 3.7 | 40.7 ± 3.5 |
| | a* | 2018 | 34.4 ± 5.6 | 36.5 ± 4.0 | 34.0 ± 6.5 | 36.9 ± 3.5 a | 33.5 ± 4.9 | 35.1 ± 4.9 |
| | | 2019 | 41.4 ± 2.6 | 42.5 ± 4.2 | 42.2 ± 1.7 | 42.6 ± 3.0 | 41.6 ± 3.2 | 42.1 ± 2.9 |
| | b* | 2018 | 17.3 ± 3.8 | 17.9 ± 3.4 a | 18.0 ± 4.6 | 18.0 ± 3.0 | 17.6 ± 4.4 | 17.8 ± 3.8 |
| | | 2019 | 21.6 ± 2.6 | 26.0 ± 4.2 | 22.6 ± 2.2 | 23.7 ± 3.6 | 21.9 ± 4.1 | 23.2 ± 3.3 |
| Aril | L* | 2018 | 24.1 ± 4.7 | 23.0 ± 7.3 | 24.6 ± 6.1 | 23.0 ± 7.3 | 23.3 ± 6.4 | 23.6 ± 6.4 |
| | | 2019 | 20.4 ± 3.1 | 22.1 ± 5.3 | 23.1 ± 4.9 | 22.5 ± 5.1 | 22.8 ± 5.9 | 22.2 ± 4.9 |
| | a* | 2018 | 18.0 ± 4.3 | 16.3 ± 5.4 | 17.9 ± 7.5 | 16.3 ± 5.4 | 16.8 ± 4.2 | 17.1 ± 5.4 |
| | | 2019 | 19.1 ± 5.0 | 16.3 ± 4.3 | 16.2 ± 6.1 | 17.1 ± 5.1 | 16.0 ± 4.0 | 16.9 ± 4.9 |
| | b* | 2018 | 6.8 ± 1.8 | 7.2 ± 2.5 | 6.9 ± 3.4 | 6.5 ± 2.6 | 6.8 ± 1.9 | 6.8 ± 2.4 |
| | | 2019 | 5.9 ± 1.3 | 6.3 ± 1.5 | 5.9 ± 1.9 | 7.0 ± 1.6 | 6.6 ± 2.0 | 6.4 ± 1.7 |
| Juice | L* | 2018 | 24.0 ± 2.7 | 21.0 ± 1.1 | 23.1 ± 4.5 | 21.4 ± 1.9 | 20.8 ± 2.1 | 22.1 ± 2.5 |
| | | 2019 | 16.8 ± 1.5 | 18.3 ± 1.0 | 16.8 ± 1.4 | 16.2 ± 1.8 | 17.8 ± 2.3 | 17.2 ± 1.6 |
| | a* | 2018 | 5.6 ± 1.4 | 6.7 ± 2.7 | 6.1 ± 2.1 | 5.1 ± 2.1 | 4.2 ± 1.4 | 5.5 ± 1.9 |
| | | 2019 | 2.2 ± 0.4 | 5.1 ± 1.0 | 2.9 ± 1.5 | 2.9 ± 1.1 | 3.9 ± 2.2 | 3.3 ± 1.2 |
| | b* | 2018 | 2.9 ± 0.5 | 2.6 ± 0.6 | 2.3 ± 0.6 | 1.3 ± 0.7 | 2.1 ± 0.5 | 3.3 ± 0.6 |
| | | 2019 | 2.2 ± 0.5 | 2.5 ± 1.1 | 1.9 ± 0.4 | 2.0 ± 0.3 | 2.2 ± 0.9 | 2.2 ± 0.6 |

Average values ± std. dev. are not statistically different from each other. Different letters per year indicate significant differences at $p < 0.05$, according to the Tukey's test.

### 3.6. Physico-Chemical Parameters of Juices

Table 6 shows the physico-chemical traits of juice under different treatments. As for TSS, pH, TA, and TSS/TA no significant differences among treatments in either season were observed; their mean values were 17.3 °Brix, 3.0, 1.6 g citric acid 100 mL$^{-1}$ and 11.2, respectively.

**Table 6.** Effect of different biostimulants on physic-chemical traits of pomegranate cv. Wonderful juices.

| Parameter | Year | Biostimulant Treatment | | | | | Average Yields |
|---|---|---|---|---|---|---|---|
| | | Control | Hendophit PS® | Ergostim XL® | Siapton 10L® | Allibio-Rad® | |
| Total soluble Solid (°Brix) | 2018 | 17.4 ± 1.3 | 17.1 ± 0.4 | 17.3 ± 0.1 | 17.4 ± 0.1 | 17.6 ± 1.1 | 17.4 ± 0.6 |
| | 2019 | 16.9 ± 0.7 | 17.4 ± 0.1 | 17.4 ± 0.1 | 16.8 ± 0.4 | 17.5 ± 0.2 | 17.2 ± 0.3 |
| pH | 2018 | 2.9 ± 0.2 | 2.9 ± 0.1 | 2.9 ± 0.1 | 2.9 ± 0.1 | 2.9 ± 0.1 | 2.9 ± 0. 1 |
| | 2019 | 3.0 ± 0.1 | 3.1 ± 0.2 | 2.9 ± 0.2 | 3.2 ± 0.2 | 3.1 ± 0.1 | 3.1 ± 0.1 |
| Total acidity (g L$^{-1}$ citric acid) | 2018 | 1.5 ± 0.1 | 1.4 ± 0.1 | 1.7 ± 0.1 | 1.6 ± 0.1 | 1.6 ± 0.1 | 1.6 ± 0.1 |
| | 2019 | 1.3 ± 0.2 | 1.5 ± 0.1 | 1.7 ± 0.2 | 1.6 ± 0.2 | 1.7 ± 0.2 | 1.6 ± 0.2 |
| Maturity index (%) | 2018 | 11.6 ± 0.6 | 12.2 ± 1.1 | 10.2 ± 0.8 | 10.9 ± 0.7 | 11.0 ± 1.2 | 11.2 ± 0.8 |
| | 2019 | 13.0 ± 0.6 | 11.6 ± 0.8 | 10.2 ± 0.8 | 10.8 ± 0.9 | 10.3 ± 0.9 | 11.2 ± 0.8 |

The data are not statistically different to each other.

Our TSS data are very similar to those reported in other studies [62,66–68], but higher than those reported by Martínez et al. [68], and slightly lower than those obtained by



Adiletta et al. [69]. Furthermore, our pH data are similar to that of other study [61,68–70] but they were slightly lower than those obtained by Fernandes et al. [68]. Finally, our MI data, according to the Martinez et al. [68] classification can be identified as "sour" or "sour–sweet".

### 3.7. Total Phenols, Antioxidant Activity

The results show that total phenol content (TPC) and antioxidant activity (AA) had higher values in 2019 than in 2018 (Figure 5). These differences may be explained by the driest weather that was recorded in 2019; this may have caused water stress in the plants and an increase in the above-mentioned juice traits. In this regard, Attanayake et al. [70] indicated that the accumulation of bioactive compounds was relatively higher in drier and warmer climates than in wetter and cooler conditions.

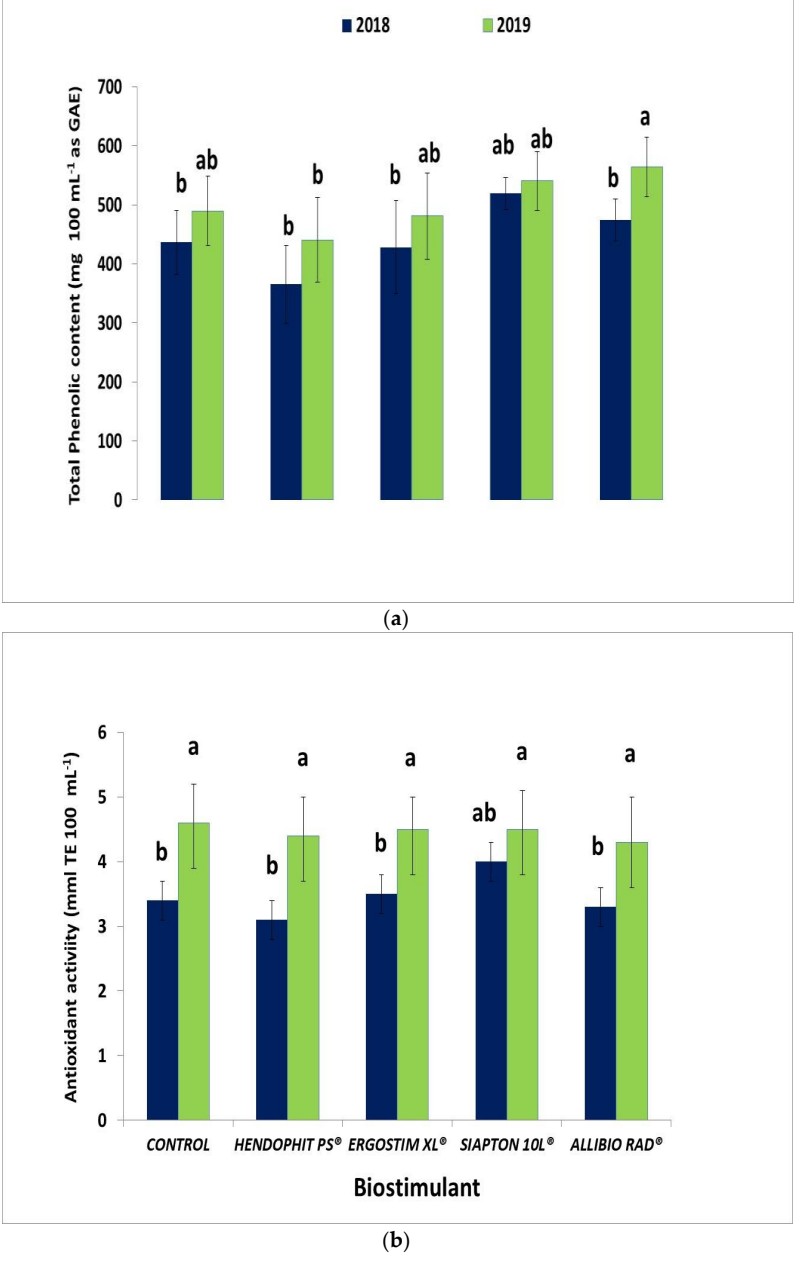

**Figure 5.** Total phenolic content (**a**) and antioxidant activity (**b**) in different biostimulant and control treatments. Average values ± sd in 2018 and 2019 are shown. Different letters per year indicate significant differences at $p < 0.05$, according to the Tukey's test.

As for the different treatments, in 2018 no significant differences were shown (values ranged from 365 and 519.3 mg 100 mL$^{-1}$), while in 2019, the highest significant TPC values were noted for Allibio-Rad® (564 mg 100 mL$^{-1}$) which was not significantly different to Siapton 10L® (540.2 mg 100 mL$^{-1}$), Ergostim XL® (481.1 mg 100 mL$^{-1}$), or the control (489.2 mg 100 mL$^{-1}$).

The TPC range values of "Wonderful" cultivar juice were similar to those reported for the same cultivar tested by Passafiume et al. [64]. Concerning antioxidant activity, except for the already mentioned significant differences between the years (average values 3.5 mmol TE 100 mL$^{-1}$ in 2018 and 4.5 mmol TE 100 mL$^{-1}$ in 2019), no foliar applications of organo-mineral fertilizers had a statistically significant effect on this parameter.

## 4. Conclusions

From the results obtained in this investigation, and despite some differences between the years, it can be inferred that the different treatments with each foliar fertilizer used—Hendophyt® PS, Ergostim® XL, Siapton10L®, and AlliBio-Rad® (containing plant biostimulating molecules such as polyglucosamine, carboxylic acids, peptides, and humic and fulvic acids, respectively)—positively influenced the tree growth, number of fruit per tree, aril number per fruit, edible portion, and juice yield of young pomegranate trees. Even though the other characteristics of the fruits were not significantly affected by the foliar application of the biostimulants, many of them showed generally better values than the control. These treatments could, therefore, be recommended to improve the performance of pomegranate cv. "Wonderful" under conditions similar to those in this study.

Additionally, further investigations are needed regarding the doses to be used for each of the products under investigation in this trial, and the number of their applications to clarify the possible significance of their effects on the numerous other parameters that have shown better values following the application of the different products.

**Author Contributions:** Conceptualization, A.T., G.D., L.F. and G.L.; methodology, A.T., G.D., L.F. and G.L.; software, A.T. and G.D.; valida-tion, A.T., G.D. and G.L.; formal analysis, A.T., G.D. and L.F.; investigation, A.T., G.D., L.F. and G.L.; data curation, A.T., G.D. and L.F.; writing—original draft preparation, A.T., G.D., L.F. and G.L.; writing—review and editing, A.T., G.D., L.F. and G.L.; visualization, A.T., G.D., L.F. and G.L.; supervision, A.T. and G.D. All authors have read and agreed to the published version of the manuscript.

**Funding:** This research received no external funding.

**Data Availability Statement:** Not applicable.

**Conflicts of Interest:** The authors declare no conflict of interest.

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
