# Peer review of "Organo Mineral Fertilizers Increases Vegetative Growth and Yield and Quality Parameters of Pomegranate cv. Wonderful Fruits"

_horticulturae, doi:10.3390/horticulturae9020164_

Round 1

Reviewer 1 Report

Dear Authors:

The following changes are necessary for the manuscript:

Abstract

Point 1. This part is not written correctly or clearly in general. The authors should describe the most noteworthy results from each year in this part, as well as the differences between years. The authors make no mention of any phyiscochemical results.

Point 2. L11: In a few paragraphs before mentioning the goal, the authors should identify the problem and explain why they chose the current technique.

Point 3. L13 (compared with control): These words should be deleted

Point 4. L14 (some physical, chemical and phytochemicals parameters): These words should be combined and changed to physicochemical parameters

Point 5. L15 (The foliar treatments were applied three times along each season (at red ball, fruit setting and fruit development stages): This sentence should be combined with the aim of study and converted into a single sentence

Point 6. L16 (The results obtained showed interaction between years and biostimulant 16 applications): This line has no sense and should be rewritten

Point 7. L26: A study conclusion should be included.

Introduction

Point 1. In general, this section is poorly written and should be revised. In this section, the authors should demonstrate how the difficulties affecting pomegranate growth and quality, particularly the effects of artificial fertilizers on the ecosystem. Furthermore, the author should describe the differences between biostimulants and synthetic fertilizers. There is no information available on the nutritional and chemical content of pomegranate.

Point 2. L31-33 (The cultivation of pomegranate (Punica granatum L.) worldwide has increased 31 sharply in the past few years, mainly due to the growing perception that this fruit has 32 numerous medical benefits): This line needs a references

Point 3. L64 (Availability of information is relatively limited for pomegranate): This line has no sense.

Point 4. L69 (Very little is known on ‘Wonderful’ pomegranate, an industry standard cultivar): This line should be deleted.

Point 5. L73: The goal is not properly phrased and should be revised.

Point 6, The authors should state the study's hypothesis before addressing the study's goal.

Point 7. Following the description of the goal, the authors should add a few sentences about how this study would benefit sustainability agriculture. Following the description of the goal, the authors should add a few sentences about how this study would benefit sustainability agriculture.

Materials and methods

Point 1. This manuscript does not include the term (Materials and methods).

Point 2. L82 (section 2.1): The climate conditions, including minimum and maximum temperatures, humidity, and light duration, should be mentioned by the authors. In addition, some information about soil's physical and chemical qualities is provided.

Point 3. L104-108 (To avoid any contamination between treatments three centrally located plants per plot were used to collect the vegetative and reproductive parameters, whereas one buffer row was used to separate plots on adjacent rows and two or more buffer rows around the perimeter of the experimental field. Trees were selected to be healthy and as uniform as possible): These lines are unclear and should be revised.

Point 4. L102-105: The space between two trees should be inserted

Point 5. L120 (The annual trunk growth (ATG) and the trunk cumulative growth (TCG) were also calculated): How did the authors compute the ATG and TCG?

Point 6. L151 (JMP® software): The version should be added

Results and discussion

Point 1. L157 (section 3.1): This section should be placed in Materials and methods

Point 2. L186 (from April 2017 to October 2018): The authors should double-check this statement, especially the year, because Figure 2 mentions different outcome.

Point 3. The Y axis of Figures 2 and 3 should be started by 0

Point 4. L219 (Table 2): The statistical data analysis should be performed for this table

Point 5. Tables 2, 3, and 5 compare different biostimulants in each year, but no comparison between two years is provided.

Point 6. L279 (3.6. Technological parameters of juices): The title of this section should be phyisco-chemical traits

Point 7. Figure 3 is not in its proper placement; relocate Figure 3.

Discussion

Point 1. There is no discussion in this manuscript. The authors should discuss the influence of each biostimulant on growth, yield, and biochemical characteristics, depending on the content of the biostimulant.

Point 2. Furthermore, the authors must interpret the effect of seasons on the characteristics under consideration.

Best regards

Author Response

Revision of the manuscript “Foliar application effect of organo mineral fertilizers with biostimulating action on the vegetative growth, yield, and quality of wonderful pomegranate fruits in two consecutive years”, by Annalisa Tarantino, Grazia Disciglio , Laura Frabboni and Giuseppe Lopriore.

Dear Editor,

We appreciate the opportunity to revise the above-mentioned manuscript submitted to Horticulturae. We sincerely thank you and the reviewers for the effort and constructive remarks.

Below we have addressed the reviewers’ comments. Additions to the manuscript have been highlighted in green in this letter, while in red in the revised manuscript.

Sincerely yours,

Dr. Annalisa Tarantino

e-mail: annalisa.tarantino@unifg.it

reviewer 1

Comments to Author

Abstract:

Point 1. Abstract. This part is not written correctly or clearly in general. The authors should describe the most noteworthy results from each year in this part, as well as the differences between years. The authors make no mention of any phyiscochemical results.

Thank you for your observation, we have modified de Abstract as following:

Abstract: “ In recent years, to improve sustainable production in horticultural crops, many new types of strategies  have been developed, including organo-mineral fertilization to complement chemical fertilizers in order to improve plant nutrients and sustainable development of agroecosystems1 The study was performed on young pomegranate trees of Wonderful cultivar, in 2018 and 2019 seasons, in order to evaluate the effects of three foliar application along each season (at red ball, fruit setting and fruit development stages) of four commercial organo-mineral fertilizers (Hendophyt®, Ergostim XL®, Siapton® 10L and Allibio Rad®) on vegetative growth, yield, and some  physico-chemical parameters of fruits. The results obtained showed some  differences between years. The annual trunk growth of trees in all compared treatments shows significantly lower values in 2018 (average 9.7 mm), than in 2019 (average 11.8 mm), At the end of the two-year period the biostimulant treatments provided significantly larger trunk diameters (average  43.6 mm) than the control (39.6 mm). Only in 2018 , significant higher  number of fruits per tree, number of arils per fruit, edible part and the juice yield were obtained in biostimulant treatments compared to  control. No differences among treatments were observed for all color parameters and physico-chemical traits of the fruits in both years. In 2019, the fruit morpho-pomological properties tend to be lower than in 2018, while on the contrary the total phenol content and the antioxidant activity were higher. Probably the warmer and windier climate of 2019 led to a condition of greater stress for the plants, with a reduction in fruit size and an increase of  bioactive compounds of juice. In conclusion, due to the various positive results, foliar organo-mineral fertilizers could be recommended to improve the performance of Wonderful pomegranate cv, under  similar conditions..”

Point 2. L11: In a few paragraphs before mentioning the goal, the authors should identify the problem and explain why they chose the current technique.

Thank you for your observation Therefore, we have added the following sentence to the abstract:

“In recent years, to improve sustainable production in horticultural crops, many new types of strategies  have been developed, including sustainable fertilization to complement chemical fertilizers in order to improve plant nutrients and development”.

Point 3. L13 (compared with control): These words should be deleted

We have deleted the words "compared with control” in L13

Point 4. L14 (some physical, chemical and phytochemicals parameters): These words should be combined and changed to physicochemical parameters

Thank you for your observation.

The words physical, chemical and phytochemicals parameters have been combined and changed to “physicochemical parameters” in L14.

Point 5. L15 (The foliar treatments were applied three times along each season (at red ball, fruit setting and fruit development stages): This sentence should be combined with the aim of study and converted into a single sentence

Thank you for your observation.

We have combined and converted the sentence as follows:

“The study was performed on young pomegranate trees of Wonderful cultivar in 2018 and 2019 seasons,  in order to evaluate the effects of three foliar application along each season (at red ball, fruit setting and fruit development stages) of four commercial organo-mineral fertilizers (Hendophyt®, Ergostim XL®, Siapton® 10L and Allibio Rad®), on vegetative growth, yield, and some  physicochemical parameters of pomegranate fruits”.

Point 6. L16 (The results obtained showed interaction between years and biostimulant 16 applications):This line has no sense and should be rewritten

Thank you for your observation.

We have rewritten the sentence as follows:

“The results obtained showed some differences between years”.

Point 7. L26: A study conclusion should be included.

Thank you for your observation In conclusion, due to the various positive results, foliar organo-mineral fertilizers could be recommended to improve the performance of Wonderful pomegranate cv, under  similar conditions.

Introduction

Point 1. In general, this section is poorly written and should be revised. In this section, the authors should demonstrate how the difficulties affecting pomegranate growth and quality, particularly the effects of artificial fertilizers on the ecosystem. Furthermore, the author should describe the differences between biostimulants and synthetic fertilizers. There is no information available on the nutritional and chemical content of pomegranate. We agree with the reviewer’s comment. Therefore, we have added many parts in the introduction, also adding the new  references (from L38 to L 49)  as follows:As for other perennial fruit tree species, also for pomegranate the fertilization is one of the important management tools in increasing growth and quanti-qualitative yield parameters. Therefore, it is  important to prevent any nutritional stress other than others due to abiotic factors (i.e. drought, soil salinity, and different climatic parameters) to which this crop can be exposed[3-5]. In this  regard,  Wonderful pomegranate cv confronts a lot of serious problems such as fruit cracking, sunburn, lack of coloration in the appropriate color of the peel and pulp,  and lowest average of yield. [Abd El-wahed et al,2021)

Abd El-wahed A. N., Abd –Alrazik A. M., and Khalifa S. M. Effect of some nutrients on growth, Yield and fruit quality of “Wonderful” cultivar pomegranate Al-Azhar Journal of Agricultural Research V. (46) No. (1) June (2021) 1-15

There are several kinds of nutrients necessary for plant growth, however, N and P are two of the most important and abundant ones. Particularly N is the primary nutrients of the crop, the requirement for which is equal to the quantity of N contained in the harvested fruit, which is approximately 0.25 kg N per 100 kg of fruit. (http://webapps.iihr.res.in>crop).  N and/or P are critical determinants of plant growth in most ecosystems. Synthetic fertilizers can harm the environment because their nitrogen and phosphorus levels are often higher. Excessive use of nitrogen can impact not only the climate but also damage plant health and reduce biodiversity both on land and in our waterways. Long-term land application of phosphorous-enriched fertilizers leads to phosphorus accumulation in soil that may become susceptible to mobilization via erosion, surface runoff and subsurface leaching ( Shama, 2021)

Shama E. Haque. How effective are existing phosphorus management strategies in mitigating surface water quality problems in the U.S.? Sustainability 202113(12), 6565; https://doi.org/10.3390/su13126565

To deal with these problems, conventional agriculture needs to increase its independence from chemical fertilizers, which have a serious impact on the natural ecosystem and human health. [

The plant production should be based on stimulating plant growth and development, while reducing risks posed to humans and the natural environment, as well as at providing safe high quality agricultural products. (Posmyk  et al. 2016)

Posmyk MM, SzafraÅ„ska K. Biostimulators: A new trend towards solving an old problem. Front. Plant Sci. 2016;7:48. doi: 10.3389/fpls.2016.00748.

Therefore, the aim of modern agriculture is to use the new sustainable technology that are dedicated to the sustainable development of agroecosystems1 Therefore, also the fertilization techniques have moved towards organic, sustainable or environmental friendly strategy,  to complement chemical fertilizers in order to improve nutrient [6,7]. In this context from some years, ‘Agricultural Biostimulants’ (ABs), defined as products stimulating plants’ natural nutrition processes,  to support crop growth, yield and to improve the final quality of products [8-10]. ABS as
natural compounds, are compatible with the plants, environment and non-toxic, compared with synthetic fertilizers  to use classic.

As for the nutritional and chemical content  of the pomegranate fruit, various data are reported in some of our previous experimental researches

Point 2. L31-33 (The cultivation of pomegranate (Punica granatum L.) worldwide has increased 31 sharply in the past few years, mainly due to the growing perception that this fruit has 32 numerous medical benefits):This line needs a references

We have added the following reference:

Chater, J.M.; Garner, L.C. Foliar nutrient applications to 'Wonderful' pomegranate (Punica granatum L.). I. Effects on fruit mineral nutrient concentrations and internal quality. Sci. Hortic. 2019, 244, 421-427.

Point 3. L64 (Availability of information is relatively limited for pomegranate): This line has no sense.

We have replaced the above sentence with the following:

The effect of foliar applied biostimulant substances on yield and fruit quality of pomegranate have been studied on several cultivars, but information on this effect on cv Wonderful  is very scarce.

Point 4. L69 (Very little is known on ‘Wonderful’ pomegranate, an industry standard cultivar): This line should be deleted.

We have deleted the above sentence

Point 5. L73: The goal is not properly phrased and should be revised.Point 6, The authors should state the study's hypothesis before addressing the study's goal.Point 7. Following the description of the goal, the authors should add a few sentences about how this study would benefit sustainability agriculture. 

We thank the Reviewer for these considerations

We have revised the goal, stated study’s hypothesis and added the sentence on the benefit  as follows:

“The aim of this study is to evaluate in a semiarid environment of Apulia region of Italy, characterized also by a high wind speed, the effects of foliar application of four commercial organo-mineral fertilizers (Hendophyt®, Ergostim XL®, Siapton® 10L and Allibio Rad®), on vegetative growth, yield, and some physicochemical parameters of Wonderful® pomegranate cultivar, in order to reduce fertilizer application rates and  enhance plants’ nutrient use efficiency. In addition,  the application of organo-mineral fertilizers as biostimulants in horticultural crops can be a particularly valuable tool in agroecological and sustainable agriculture”.

Materials and methods

Point 1. This manuscript does not include the term (Materials and methods).

The authors  apologize to the reviewer for the mistake.

We have inserted the termMaterials and methods”

Point 2. L82 (section 2.1): The climate conditions, including minimum and maximum temperatures, humidity, and light duration, should be mentioned by the authors. In addition, some information aboutsoil's physical and chemical qualities is provided.

We have inserted  in the manuscript the table 1 of climatic parameters and the soil physicochemical parameters as follows:

                                                   Table 1 - Monthly mean maximum and minimum temperatures (Tmax, Tmin) and

relative air humidity (RHma  and RHmin), wind speed (Ws), radiation (Rad) and total precipitation (P) in 2018 and 2019 seasons.

Month

Tmax

Tmin

RHmax

RHmin

Ws

Rad

P

(°C)

(°C)

(%)

(%)

(m sec-1)

(Wm-2)

(mm)

2018

April

21.3

12.9

94.6

37.6

2.8

235.3

54.0

May

26.1

13.4

95.2

49.1

2.4

275.8

58.3

June

30.0

12.1

89.5

40.3

3.4

289.6

88.2

July

33.3

19.6

83.6

35.4

3.0

318.7

16.8

Aug

32.7

20.1

71.3

28.3

2.1

285.7

39.1

Sept

29.1

17.1

81.3

30.0

3.7

193.6

80.0

Mean

23.7

15.9

85.9

36.8

2.9

266.5

Total

366.4

2019

April

20.6

8.2

94.4

51.0

3.7

190.2

40.3

May

21.3

10.2

95.3

56.3

4.0

232.9

86.7

June

33.2

17.5

85.9

35.1

3.7

252.2

9.2

July

33.7

19.5

84.0

33.9

3.7

258.8

30.0

Aug

34.8

20.3

79.9

33.9

3.6

225.6

5.7

Sept

29.5

16.8

88.7

42.6

3.6

175.5

3.8

Mean

28.8

15.4

88.0

42.5

3.7

222.5

Total

175.7

 The soil is a silty-clay vertisol of alluvial origin (1.20 m depth) (Typic Chromoxerert, fine, thermic, according to the Soil Taxonomy-USDA), with the following characteristics: 32.7% clay; 30.5% silt; 36.8 % sand; pH 8.8 (soil:water 1:2.5); 1.4% organic matter; 1.5‰ total N, 9.7% active CaCO3. 56 mg kg–1 P205 and 1390 mg kg–1 exchageable K.

Point 3. L104-108 (To avoid any contamination between treatments three centrally located plants per plot were used to collect the vegetative and reproductive parameters, whereas one buffer row was used to separate plots on adjacent rows and two or more buffer rows around the perimeter of the experimental field. Trees were selected to be healthy and as uniform as possible): These lines are unclear and should be revised.

Thank you for your observation

We revised the above sentence as follows:

“To avoid any contamination between treatment one buffer row was located between replicates and blocks along two or more buffer rows around the perimeter of the experimental field.  In each replicate three centrally located plants per plot were used to collect vegetative and reproductive parameters.

Point 4. L102-105: The space between two trees should be inserted

Thank you for your observation

We have inserted the following sentence:“Trees were planted at a distance of 3 m on the row and 5.5 m between the rows”

Point 5. L120 (The annual trunk growth (ATG) and the trunk cumulative growth (TCG) were also calculated): How did the authors compute the ATG and TCG?

 We have inserted the following sentence:The ATG was calculated, by making the difference between the measurement made in October and that in April  each year, whereas TCG was calculated, by making the difference between the measurement made on October 25th, 2019 and that on April 5th, 2018. 

Point 6. L151 (JMP® software): The version should be added

We added:

“package, version 14.3” 

Results and discussion

Point 1. L157 (section 3.1): This section should be placed in Materials and methods

We have placed the climate paragraph in the Materials and methods

Point 2. L186 (from April 2017 to October 2018): The authors should double-check this statement,

especially the year, because Figure 2 mentions different outcome.

The authors  apologize to the reviewer for the mistake.

We have corrected the dates in the figure 2  caption.

Point 3. The Y axis of Figures 2 3 and 3 4 should be started by 0

Figures 3 and 4 have been corrected in the Y axes, starting from 0  

Point 4. L219 (Table 2): The statistical data analysis should be performed for this table

Thank you for your observation We have inserted  the caption under the table 2 and the following sentence in the manuscript: “All the data are not statistically different among them”.“The average weight, diameter and length of the fruits in general tend to be lower in 2019 than in 2018, even if the data are not statistically different from each other”.

Point 5. Tables 2, 3, and 5 compare different biostimulants in each year, but no comparison between two years is provided.

The authors warmly tank the reviewer for the proposed suggestion

We have correct the tables for the comparison  between two years as follows:

Table 3 - Effect of different biostimulants on morpho-pomological attributes of ‘Wonderful’ pomegranate whole fruits

Parameter

Year

Biostimulant treatment

Average years

Control

Hendophit PS®

Ergostim XL®

Siapton 10L®

Allibio-rad®

Fruit average weight (g)

2018

411.1±72.8

457.0±142.0

512.1±106.2

476.0±119.1

473.6±124.6

465.96±112.94

2019

414.4 ±87.3

409.3±69.4

440.0±74.6

477.1±101.4

393.4±59.8

426.84±78.5

Fruit diameter (mm)

2018

86.5±12.2

91.0±10.6

89.5±12.6

93.2±9.3

86.8±13.4

89.4±11.62

2019

85.6± 10.2

87.5±8.2

90.8±6.5

89.9±6.6

84.8±6.6

87.72±7.62

Fruit length (mm)

2018

74.4±10.6

81.9±10.2

80.4±9.1

82.9±9.3

80.3±9.1

79.98±9.66

2019

74.6±12.2

76.6±8.5

80.2±6.8

81.9±8.6

75.3±7.2

77.72±8.66

Table 4 - Effect of different biostimulants on the main constituent parts of pomegranate fruits of the cv. ‘Wonderful’.

Parameter

Year

Biostimulant treatment

Average years

Control

Hendophit PS®

Ergostim XL®

Siapton 10L®

Allibio-rad®

Aril weight per fruit (g)

2018

210.8±57.0

234.6±81.4

263.0±86.8

223.8±71.5

230.9±70.3

193.8±73.4

2019

184.2±47.6

184.5± 61.3

207.7±44.0

217.8±63.0

168.2±59.6

160.4±55.1

Fresh weight 100 arils (g)

2018

28.6±4.4

28.8±5.4

30.2±3.1

29.0±3.7

29.7±4.7

24.4±4.3

2019

24.5±1.7

28.7±4.7

29.8±1.2

24.8±1.8

24.9±4.3

22.1±2.7

Aril per fruit (No)

2018

734.5±19.6 b

814.6±25.9 a

870.9±28.6 a

771.7±23.4 b

864.8±25.0 a

676.1±24.5 a

2019

721.4±16.1c

691.2±15,3c

688.1±18.4c

761.1±14.6 b

740.2±16.1b

600.3±14.1 b

Fruit edible portion (%)

2018

48.2±1.0 b 

51.3±1.4 a

51.3±1.8 a

49.5±1.4 ab

48.9±1.3 a b

41.5±1.38 a

2019

44.3±4.0 b

48.5±7.4ab

46.6±5.0ab

45.6± 3.9 ab

46.9±4.3 ab

38.7±4.92 b

Juice volume (cm3 100 g-1)

2018

81.1±2.3 cd

100.8±2.9 a

96.2±2.3ab

94.2±2.3 b

95.6±2.2ab

78.0±2.4 a

2019

77.7±2.1cd

97.3±2.0 ab

98.1±1.7ab

82.3±2.1c

81.3±1.9 cd

72.8±1.96 b

Dry matter arils (%)

2018

20.1±0.4

20.3±0.5

20.4±0.4

20.5±0.5

20.3±0.5

16.9±0.46

2019

23.3±1.0

22.3±0.8

21.1±1.1

22.1±0.9

21.4±0.9

18.4±0.94

Table 5 - Effect of different biostimulants on skin, aril and juice color characteristics of ‘Wonderful’ pomegranate fruits

Parameter

Year

Biostimulant treatment

Average years

Control

Hendophit PS®

Ergostim XL®

Siapton 10L®

Allibio-rad®

Skin

L*

2018

42.1±6.0

42.3±5.5

43.1±6.7

42.3±5.3

42.2±6.2

42.4±5.9

2019

40.0±2.6

42.6±5.6

40.0±2.1

40.2±3.7

40.5±3.7

40.7±3.5

a*

2018

34.4±5.6

36.5 ±4.0

34.0±6.5

36.9±3.5 a

33.5±4.9

35.1±4.9

2019

41.4 ±2.6

42.5±4.2

42.2±1.7

42.6±3.0

41.6 ±3.2

42.1±2.9

b*

2018

17.3±3.8

17.9±3.4 a

18.0±4.6

18.0±3.0

17.6±4.4

17.8±3.8

2019

21.6±2.6

26.0±4.2

22.6±2.2

23.7±3.6

21.9±4.1

23.2±3.3

Aril

L*

2018

24.1±4.7

23.0±7.3

24.6±6.1

23.0±7.3

23.3±6.4

23.6±6.4

2019

20.4± 3.1

22.1±5.3

23.1±4.9

22.5±5.1

22.8±5.9

22.2±4.9

a*

2018

18.0±4.3

16.3±5.4

17.9±7.5

16.3±5.4

16.8±4.2

17.1±5.4

2019

19.1±5.0

16.3±4.3

16.2±6.1

17.1±5.1

16.0±4.0

16.9±4.9

b*

2018

6.8±1.8

7.2±2.5

6.9±3.4

6.5±2.6

6.8±1.9

6.8±2.4

2019

5.9±1.3

6.3±1.5

5.9±1.9

7.0±1.6

6.6±2.0

6.4±1.7

Juice

L*

2018

24.0±2.7

21.0±1.1

23.1±4.5

21.4±1.9

20.8±2.1

22.1±2.5

2019

16.8±1.5

18.3±1.0

16.8±1.4

16.2±1.8

17.8±2.3

17.2±1.6

a*

2018

5.6±1.4

6.7±2.7

6.1±2.1

5.1±2.1

4.2 ±1.4

5.5±1.9

2019

2.2±0.4

5.1±1.0

2.9± 1.5

2.9±1.1

3.9±2.2

3.3±1.2

b*

2018

2.9±0.5

2.6±0.6

2.3±0.6

1.3±0.7

2.1± 0.5

3.3±0.6

2019

2.2± 0.5

2.5±1.1

1.9±0.4

2.0±0.3

2.2±0.9

2.2±0.6

Table 6 - Effect of different biostimulants on technological parameters of ‘Wonderful’ pomegranate juices.

Parameter

Year

Biostimulant treatment

Average years

Control

Hendophit PS®

Ergostim XL®

Siapton 10L®

Allibio-rad®

Total soluble Solid (°Brix)

2018

2012018

1717.4±1.3

2018

2019

17.4±1.3

16.9±0.7

17.1±0.4

17.4±0.1

17.3±0.1

17.4±0.1

17.4±0.1

16.8± 0.4

17.6±1.1

17.5 ±0.2

17.4±0.6

17.2±0.3

pH

2018

2019

2.9±0.2

3.0 0.1

2.9±0.1

3.1 0.2

2.9±0.1

2.9 0.2

2.9± 0.1

3.2 0.2

2.9±0.1

3.1 0.1

29±0.1

3.1± 0.1

Total acidity

(g L-1 citric acid)

2018

1.5±0.1

1.4±0.1

1.7±0.1

1.6±0.1

1.6±0.1

1.6±0,1

2019

1.3±0.2

1.5 ±0.1

1.7±0.2

1.6 ±0.2

1.7± 0.2

1.6±0.2

Maturity index (%)

2018

11.6±0.6

12.2±1.1

10.2±0.8

10.9±0.7

11.0±1.2

11.2±0.9

2019

2019

13.0±0.6

11.6±0.8

10.2±0.8

10.8±0.9

10.3±0.9

11,2±0.8

Point 6. L279 (3.6. Technological parameters of juices): The title of this section should be phyisco-chemical traits

Thank you for your observation

We have correct the title with phyisco-chemical traits

Point 7. Figure 3 is not in its proper placement; relocate Figure 3.

 Thank you for your observations.

The requested replacement have been made

Discussion

Point 1. There is no discussion in this manuscript. The authors should discuss the influence of each biostimulant on growth, yield, and biochemical characteristics, depending on the content of the biostimulant.

We agree with the reviewer’s comment . In the manuscript, however, the discussion is inserted in the Results and discussion.

In this regard, we have added some sentences and new references as follows:

Indeed, he used foliar fertilizers, containing  macromolecular compounds, in addition to the typical chemical elements (carbon, oxygen, hydrogen, nitrogen, sulfur), influenced positively the tree growth as they enhance metabolism and activity of plant photosynthesis L 305 (Mihalache, 2017)

Generally, the best results obtained from the application of above biostimulants in 2019, although their nature is diverse, are to be ascribed to the more stressful climatic conditions that occurred in this season, confirming their action for mitigating the adverse impacts of harsh environmental conditions on plants (Bhupenchandra). L 214  

Mihalache D., Sirbu C., Grigore A ,Stănescu A.M., Calciu I.C. , Marin N., 2017. Physical, chemical and agrochemical characterization of some organo-mineral fertilizers Romanian Biotechnological Letters, 22, 1, 12259-12666.

Point 2. Furthermore, the authors must interpret the effect of seasons on the characteristics under

consideration.

We agree with the reviewer’s comment

We have added the following sentences

“The difference in climatic conditions over the years has influenced the various production parameters and fruit quality. In particular, in the second year of the trial, characterized by higher maximum temperatures, greater windiness and drier conditions, the size of the fruit and the individual constituents of the fruit itself tended to be smaller, on the contrary, however, the physical-chemical parameters were more higher than in 2018. These results have also been confirmed previously by other studies on pomegranate when extreme hot and dry climate during the period of maximal growth rate was associated with restricted growth and a high proportion of small-size fruit. (Joshi et al.,2021

Joshi M., Ze’ev Schmilovitch Z. and  Ginzberg I.  Pomegranate Fruit Growth and Skin Characteristics in Hot and Dry Climate Front. Plant Sci., 19 August 2021 https://doi.org/10.3389/fpls.2021.725479

Best regards

Annalisa Tarantino

Revision of the manuscript “Foliar application effect of organo mineral fertilizers with biostimulating action on the vegetative growth, yield, and quality of wonderful pomegranate fruits in two consecutive years”, by Annalisa Tarantino, Grazia Disciglio , Laura Frabboni and Giuseppe Lopriore.

Dear Editor,

We appreciate the opportunity to revise the above-mentioned manuscript submitted to Horticulturae. We sincerely thank you and the reviewers for the effort and constructive remarks.

Below we have addressed the reviewers’ comments. Additions to the manuscript have been highlighted in green in this letter, while in red in the revised manuscript.

Sincerely yours,

Dr. Annalisa Tarantino

e-mail: annalisa.tarantino@unifg.it

reviewer 1

Comments to Author

Abstract:

Point 1. Abstract. This part is not written correctly or clearly in general. The authors should describe the most noteworthy results from each year in this part, as well as the differences between years. The authors make no mention of any phyiscochemical results.

Thank you for your observation, we have modified de Abstract as following:

Abstract: “ In recent years, to improve sustainable production in horticultural crops, many new types of strategies  have been developed, including organo-mineral fertilization to complement chemical fertilizers in order to improve plant nutrients and sustainable development of agroecosystems1 The study was performed on young pomegranate trees of Wonderful cultivar, in 2018 and 2019 seasons, in order to evaluate the effects of three foliar application along each season (at red ball, fruit setting and fruit development stages) of four commercial organo-mineral fertilizers (Hendophyt®, Ergostim XL®, Siapton® 10L and Allibio Rad®) on vegetative growth, yield, and some  physico-chemical parameters of fruits. The results obtained showed some  differences between years. The annual trunk growth of trees in all compared treatments shows significantly lower values in 2018 (average 9.7 mm), than in 2019 (average 11.8 mm), At the end of the two-year period the biostimulant treatments provided significantly larger trunk diameters (average  43.6 mm) than the control (39.6 mm). Only in 2018 , significant higher  number of fruits per tree, number of arils per fruit, edible part and the juice yield were obtained in biostimulant treatments compared to  control. No differences among treatments were observed for all color parameters and physico-chemical traits of the fruits in both years. In 2019, the fruit morpho-pomological properties tend to be lower than in 2018, while on the contrary the total phenol content and the antioxidant activity were higher. Probably the warmer and windier climate of 2019 led to a condition of greater stress for the plants, with a reduction in fruit size and an increase of  bioactive compounds of juice. In conclusion, due to the various positive results, foliar organo-mineral fertilizers could be recommended to improve the performance of Wonderful pomegranate cv, under  similar conditions..”

Point 2. L11: In a few paragraphs before mentioning the goal, the authors should identify the problem and explain why they chose the current technique.

Thank you for your observation Therefore, we have added the following sentence to the abstract:

“In recent years, to improve sustainable production in horticultural crops, many new types of strategies  have been developed, including sustainable fertilization to complement chemical fertilizers in order to improve plant nutrients and development”.

Point 3. L13 (compared with control): These words should be deleted

We have deleted the words "compared with control” in L13

Point 4. L14 (some physical, chemical and phytochemicals parameters): These words should be combined and changed to physicochemical parameters

Thank you for your observation.

The words physical, chemical and phytochemicals parameters have been combined and changed to “physicochemical parameters” in L14.

Point 5. L15 (The foliar treatments were applied three times along each season (at red ball, fruit setting and fruit development stages): This sentence should be combined with the aim of study and converted into a single sentence

Thank you for your observation.

We have combined and converted the sentence as follows:

“The study was performed on young pomegranate trees of Wonderful cultivar in 2018 and 2019 seasons,  in order to evaluate the effects of three foliar application along each season (at red ball, fruit setting and fruit development stages) of four commercial organo-mineral fertilizers (Hendophyt®, Ergostim XL®, Siapton® 10L and Allibio Rad®), on vegetative growth, yield, and some  physicochemical parameters of pomegranate fruits”.

Point 6. L16 (The results obtained showed interaction between years and biostimulant 16 applications):This line has no sense and should be rewritten

Thank you for your observation.

We have rewritten the sentence as follows:

“The results obtained showed some differences between years”.

Point 7. L26: A study conclusion should be included.

Thank you for your observation In conclusion, due to the various positive results, foliar organo-mineral fertilizers could be recommended to improve the performance of Wonderful pomegranate cv, under  similar conditions.

Introduction

Point 1. In general, this section is poorly written and should be revised. In this section, the authors should demonstrate how the difficulties affecting pomegranate growth and quality, particularly the effects of artificial fertilizers on the ecosystem. Furthermore, the author should describe the differences between biostimulants and synthetic fertilizers. There is no information available on the nutritional and chemical content of pomegranate. We agree with the reviewer’s comment. Therefore, we have added many parts in the introduction, also adding the new  references (from L38 to L 49)  as follows:As for other perennial fruit tree species, also for pomegranate the fertilization is one of the important management tools in increasing growth and quanti-qualitative yield parameters. Therefore, it is  important to prevent any nutritional stress other than others due to abiotic factors (i.e. drought, soil salinity, and different climatic parameters) to which this crop can be exposed[3-5]. In this  regard,  Wonderful pomegranate cv confronts a lot of serious problems such as fruit cracking, sunburn, lack of coloration in the appropriate color of the peel and pulp,  and lowest average of yield. [Abd El-wahed et al,2021)

Abd El-wahed A. N., Abd –Alrazik A. M., and Khalifa S. M. Effect of some nutrients on growth, Yield and fruit quality of “Wonderful” cultivar pomegranate Al-Azhar Journal of Agricultural Research V. (46) No. (1) June (2021) 1-15

There are several kinds of nutrients necessary for plant growth, however, N and P are two of the most important and abundant ones. Particularly N is the primary nutrients of the crop, the requirement for which is equal to the quantity of N contained in the harvested fruit, which is approximately 0.25 kg N per 100 kg of fruit. (http://webapps.iihr.res.in>crop).  N and/or P are critical determinants of plant growth in most ecosystems. Synthetic fertilizers can harm the environment because their nitrogen and phosphorus levels are often higher. Excessive use of nitrogen can impact not only the climate but also damage plant health and reduce biodiversity both on land and in our waterways. Long-term land application of phosphorous-enriched fertilizers leads to phosphorus accumulation in soil that may become susceptible to mobilization via erosion, surface runoff and subsurface leaching ( Shama, 2021)

Shama E. Haque. How effective are existing phosphorus management strategies in mitigating surface water quality problems in the U.S.? Sustainability 202113(12), 6565; https://doi.org/10.3390/su13126565

To deal with these problems, conventional agriculture needs to increase its independence from chemical fertilizers, which have a serious impact on the natural ecosystem and human health. [

The plant production should be based on stimulating plant growth and development, while reducing risks posed to humans and the natural environment, as well as at providing safe high quality agricultural products. (Posmyk  et al. 2016)

Posmyk MM, SzafraÅ„ska K. Biostimulators: A new trend towards solving an old problem. Front. Plant Sci. 2016;7:48. doi: 10.3389/fpls.2016.00748.

Therefore, the aim of modern agriculture is to use the new sustainable technology that are dedicated to the sustainable development of agroecosystems1 Therefore, also the fertilization techniques have moved towards organic, sustainable or environmental friendly strategy,  to complement chemical fertilizers in order to improve nutrient [6,7]. In this context from some years, ‘Agricultural Biostimulants’ (ABs), defined as products stimulating plants’ natural nutrition processes,  to support crop growth, yield and to improve the final quality of products [8-10]. ABS as
natural compounds, are compatible with the plants, environment and non-toxic, compared with synthetic fertilizers  to use classic.

As for the nutritional and chemical content  of the pomegranate fruit, various data are reported in some of our previous experimental researches

Point 2. L31-33 (The cultivation of pomegranate (Punica granatum L.) worldwide has increased 31 sharply in the past few years, mainly due to the growing perception that this fruit has 32 numerous medical benefits):This line needs a references

We have added the following reference:

Chater, J.M.; Garner, L.C. Foliar nutrient applications to 'Wonderful' pomegranate (Punica granatum L.). I. Effects on fruit mineral nutrient concentrations and internal quality. Sci. Hortic. 2019, 244, 421-427.

Point 3. L64 (Availability of information is relatively limited for pomegranate): This line has no sense.

We have replaced the above sentence with the following:

The effect of foliar applied biostimulant substances on yield and fruit quality of pomegranate have been studied on several cultivars, but information on this effect on cv Wonderful  is very scarce.

Point 4. L69 (Very little is known on ‘Wonderful’ pomegranate, an industry standard cultivar): This line should be deleted.

We have deleted the above sentence

Point 5. L73: The goal is not properly phrased and should be revised.Point 6, The authors should state the study's hypothesis before addressing the study's goal.Point 7. Following the description of the goal, the authors should add a few sentences about how this study would benefit sustainability agriculture. 

We thank the Reviewer for these considerations

We have revised the goal, stated study’s hypothesis and added the sentence on the benefit  as follows:

“The aim of this study is to evaluate in a semiarid environment of Apulia region of Italy, characterized also by a high wind speed, the effects of foliar application of four commercial organo-mineral fertilizers (Hendophyt®, Ergostim XL®, Siapton® 10L and Allibio Rad®), on vegetative growth, yield, and some physicochemical parameters of Wonderful® pomegranate cultivar, in order to reduce fertilizer application rates and  enhance plants’ nutrient use efficiency. In addition,  the application of organo-mineral fertilizers as biostimulants in horticultural crops can be a particularly valuable tool in agroecological and sustainable agriculture”.

Materials and methods

Point 1. This manuscript does not include the term (Materials and methods).

The authors  apologize to the reviewer for the mistake.

We have inserted the termMaterials and methods”

Point 2. L82 (section 2.1): The climate conditions, including minimum and maximum temperatures, humidity, and light duration, should be mentioned by the authors. In addition, some information aboutsoil's physical and chemical qualities is provided.

We have inserted  in the manuscript the table 1 of climatic parameters and the soil physicochemical parameters as follows:

                                                   Table 1 - Monthly mean maximum and minimum temperatures (Tmax, Tmin) and

relative air humidity (RHma  and RHmin), wind speed (Ws), radiation (Rad) and total precipitation (P) in 2018 and 2019 seasons.

Month

Tmax

Tmin

RHmax

RHmin

Ws

Rad

P

(°C)

(°C)

(%)

(%)

(m sec-1)

(Wm-2)

(mm)

2018

April

21.3

12.9

94.6

37.6

2.8

235.3

54.0

May

26.1

13.4

95.2

49.1

2.4

275.8

58.3

June

30.0

12.1

89.5

40.3

3.4

289.6

88.2

July

33.3

19.6

83.6

35.4

3.0

318.7

16.8

Aug

32.7

20.1

71.3

28.3

2.1

285.7

39.1

Sept

29.1

17.1

81.3

30.0

3.7

193.6

80.0

Mean

23.7

15.9

85.9

36.8

2.9

266.5

Total

366.4

2019

April

20.6

8.2

94.4

51.0

3.7

190.2

40.3

May

21.3

10.2

95.3

56.3

4.0

232.9

86.7

June

33.2

17.5

85.9

35.1

3.7

252.2

9.2

July

33.7

19.5

84.0

33.9

3.7

258.8

30.0

Aug

34.8

20.3

79.9

33.9

3.6

225.6

5.7

Sept

29.5

16.8

88.7

42.6

3.6

175.5

3.8

Mean

28.8

15.4

88.0

42.5

3.7

222.5

Total

175.7

 The soil is a silty-clay vertisol of alluvial origin (1.20 m depth) (Typic Chromoxerert, fine, thermic, according to the Soil Taxonomy-USDA), with the following characteristics: 32.7% clay; 30.5% silt; 36.8 % sand; pH 8.8 (soil:water 1:2.5); 1.4% organic matter; 1.5‰ total N, 9.7% active CaCO3. 56 mg kg–1 P205 and 1390 mg kg–1 exchageable K.

Point 3. L104-108 (To avoid any contamination between treatments three centrally located plants per plot were used to collect the vegetative and reproductive parameters, whereas one buffer row was used to separate plots on adjacent rows and two or more buffer rows around the perimeter of the experimental field. Trees were selected to be healthy and as uniform as possible): These lines are unclear and should be revised.

Thank you for your observation

We revised the above sentence as follows:

“To avoid any contamination between treatment one buffer row was located between replicates and blocks along two or more buffer rows around the perimeter of the experimental field.  In each replicate three centrally located plants per plot were used to collect vegetative and reproductive parameters.

Point 4. L102-105: The space between two trees should be inserted

Thank you for your observation

We have inserted the following sentence:“Trees were planted at a distance of 3 m on the row and 5.5 m between the rows”

Point 5. L120 (The annual trunk growth (ATG) and the trunk cumulative growth (TCG) were also calculated): How did the authors compute the ATG and TCG?

 We have inserted the following sentence:The ATG was calculated, by making the difference between the measurement made in October and that in April  each year, whereas TCG was calculated, by making the difference between the measurement made on October 25th, 2019 and that on April 5th, 2018. 

Point 6. L151 (JMP® software): The version should be added

We added:

“package, version 14.3” 

Results and discussion

Point 1. L157 (section 3.1): This section should be placed in Materials and methods

We have placed the climate paragraph in the Materials and methods

Point 2. L186 (from April 2017 to October 2018): The authors should double-check this statement,

especially the year, because Figure 2 mentions different outcome.

The authors  apologize to the reviewer for the mistake.

We have corrected the dates in the figure 2  caption.

Point 3. The Y axis of Figures 2 3 and 3 4 should be started by 0

Figures 3 and 4 have been corrected in the Y axes, starting from 0  

Point 4. L219 (Table 2): The statistical data analysis should be performed for this table

Thank you for your observation We have inserted  the caption under the table 2 and the following sentence in the manuscript: “All the data are not statistically different among them”.“The average weight, diameter and length of the fruits in general tend to be lower in 2019 than in 2018, even if the data are not statistically different from each other”.

Point 5. Tables 2, 3, and 5 compare different biostimulants in each year, but no comparison between two years is provided.

The authors warmly tank the reviewer for the proposed suggestion

We have correct the tables for the comparison  between two years as follows:

Table 3 - Effect of different biostimulants on morpho-pomological attributes of ‘Wonderful’ pomegranate whole fruits

Parameter

Year

Biostimulant treatment

Average years

Control

Hendophit PS®

Ergostim XL®

Siapton 10L®

Allibio-rad®

Fruit average weight (g)

2018

411.1±72.8

457.0±142.0

512.1±106.2

476.0±119.1

473.6±124.6

465.96±112.94

2019

414.4 ±87.3

409.3±69.4

440.0±74.6

477.1±101.4

393.4±59.8

426.84±78.5

Fruit diameter (mm)

2018

86.5±12.2

91.0±10.6

89.5±12.6

93.2±9.3

86.8±13.4

89.4±11.62

2019

85.6± 10.2

87.5±8.2

90.8±6.5

89.9±6.6

84.8±6.6

87.72±7.62

Fruit length (mm)

2018

74.4±10.6

81.9±10.2

80.4±9.1

82.9±9.3

80.3±9.1

79.98±9.66

2019

74.6±12.2

76.6±8.5

80.2±6.8

81.9±8.6

75.3±7.2

77.72±8.66

Table 4 - Effect of different biostimulants on the main constituent parts of pomegranate fruits of the cv. ‘Wonderful’.

Parameter

Year

Biostimulant treatment

Average years

Control

Hendophit PS®

Ergostim XL®

Siapton 10L®

Allibio-rad®

Aril weight per fruit (g)

2018

210.8±57.0

234.6±81.4

263.0±86.8

223.8±71.5

230.9±70.3

193.8±73.4

2019

184.2±47.6

184.5± 61.3

207.7±44.0

217.8±63.0

168.2±59.6

160.4±55.1

Fresh weight 100 arils (g)

2018

28.6±4.4

28.8±5.4

30.2±3.1

29.0±3.7

29.7±4.7

24.4±4.3

2019

24.5±1.7

28.7±4.7

29.8±1.2

24.8±1.8

24.9±4.3

22.1±2.7

Aril per fruit (No)

2018

734.5±19.6 b

814.6±25.9 a

870.9±28.6 a

771.7±23.4 b

864.8±25.0 a

676.1±24.5 a

2019

721.4±16.1c

691.2±15,3c

688.1±18.4c

761.1±14.6 b

740.2±16.1b

600.3±14.1 b

Fruit edible portion (%)

2018

48.2±1.0 b 

51.3±1.4 a

51.3±1.8 a

49.5±1.4 ab

48.9±1.3 a b

41.5±1.38 a

2019

44.3±4.0 b

48.5±7.4ab

46.6±5.0ab

45.6± 3.9 ab

46.9±4.3 ab

38.7±4.92 b

Juice volume (cm3 100 g-1)

2018

81.1±2.3 cd

100.8±2.9 a

96.2±2.3ab

94.2±2.3 b

95.6±2.2ab

78.0±2.4 a

2019

77.7±2.1cd

97.3±2.0 ab

98.1±1.7ab

82.3±2.1c

81.3±1.9 cd

72.8±1.96 b

Dry matter arils (%)

2018

20.1±0.4

20.3±0.5

20.4±0.4

20.5±0.5

20.3±0.5

16.9±0.46

2019

23.3±1.0

22.3±0.8

21.1±1.1

22.1±0.9

21.4±0.9

18.4±0.94

Table 5 - Effect of different biostimulants on skin, aril and juice color characteristics of ‘Wonderful’ pomegranate fruits

Parameter

Year

Biostimulant treatment

Average years

Control

Hendophit PS®

Ergostim XL®

Siapton 10L®

Allibio-rad®

Skin

L*

2018

42.1±6.0

42.3±5.5

43.1±6.7

42.3±5.3

42.2±6.2

42.4±5.9

2019

40.0±2.6

42.6±5.6

40.0±2.1

40.2±3.7

40.5±3.7

40.7±3.5

a*

2018

34.4±5.6

36.5 ±4.0

34.0±6.5

36.9±3.5 a

33.5±4.9

35.1±4.9

2019

41.4 ±2.6

42.5±4.2

42.2±1.7

42.6±3.0

41.6 ±3.2

42.1±2.9

b*

2018

17.3±3.8

17.9±3.4 a

18.0±4.6

18.0±3.0

17.6±4.4

17.8±3.8

2019

21.6±2.6

26.0±4.2

22.6±2.2

23.7±3.6

21.9±4.1

23.2±3.3

Aril

L*

2018

24.1±4.7

23.0±7.3

24.6±6.1

23.0±7.3

23.3±6.4

23.6±6.4

2019

20.4± 3.1

22.1±5.3

23.1±4.9

22.5±5.1

22.8±5.9

22.2±4.9

a*

2018

18.0±4.3

16.3±5.4

17.9±7.5

16.3±5.4

16.8±4.2

17.1±5.4

2019

19.1±5.0

16.3±4.3

16.2±6.1

17.1±5.1

16.0±4.0

16.9±4.9

b*

2018

6.8±1.8

7.2±2.5

6.9±3.4

6.5±2.6

6.8±1.9

6.8±2.4

2019

5.9±1.3

6.3±1.5

5.9±1.9

7.0±1.6

6.6±2.0

6.4±1.7

Juice

L*

2018

24.0±2.7

21.0±1.1

23.1±4.5

21.4±1.9

20.8±2.1

22.1±2.5

2019

16.8±1.5

18.3±1.0

16.8±1.4

16.2±1.8

17.8±2.3

17.2±1.6

a*

2018

5.6±1.4

6.7±2.7

6.1±2.1

5.1±2.1

4.2 ±1.4

5.5±1.9

2019

2.2±0.4

5.1±1.0

2.9± 1.5

2.9±1.1

3.9±2.2

3.3±1.2

b*

2018

2.9±0.5

2.6±0.6

2.3±0.6

1.3±0.7

2.1± 0.5

3.3±0.6

2019

2.2± 0.5

2.5±1.1

1.9±0.4

2.0±0.3

2.2±0.9

2.2±0.6

Table 6 - Effect of different biostimulants on technological parameters of ‘Wonderful’ pomegranate juices.

Parameter

Year

Biostimulant treatment

Average years

Control

Hendophit PS®

Ergostim XL®

Siapton 10L®

Allibio-rad®

Total soluble Solid (°Brix)

2018

2012018

1717.4±1.3

2018

2019

17.4±1.3

16.9±0.7

17.1±0.4

17.4±0.1

17.3±0.1

17.4±0.1

17.4±0.1

16.8± 0.4

17.6±1.1

17.5 ±0.2

17.4±0.6

17.2±0.3

pH

2018

2019

2.9±0.2

3.0 0.1

2.9±0.1

3.1 0.2

2.9±0.1

2.9 0.2

2.9± 0.1

3.2 0.2

2.9±0.1

3.1 0.1

29±0.1

3.1± 0.1

Total acidity

(g L-1 citric acid)

2018

1.5±0.1

1.4±0.1

1.7±0.1

1.6±0.1

1.6±0.1

1.6±0,1

2019

1.3±0.2

1.5 ±0.1

1.7±0.2

1.6 ±0.2

1.7± 0.2

1.6±0.2

Maturity index (%)

2018

11.6±0.6

12.2±1.1

10.2±0.8

10.9±0.7

11.0±1.2

11.2±0.9

2019

2019

13.0±0.6

11.6±0.8

10.2±0.8

10.8±0.9

10.3±0.9

11,2±0.8

Point 6. L279 (3.6. Technological parameters of juices): The title of this section should be phyisco-chemical traits

Thank you for your observation

We have correct the title with phyisco-chemical traits

Point 7. Figure 3 is not in its proper placement; relocate Figure 3.

 Thank you for your observations.

The requested replacement have been made

Discussion

Point 1. There is no discussion in this manuscript. The authors should discuss the influence of each biostimulant on growth, yield, and biochemical characteristics, depending on the content of the biostimulant.

We agree with the reviewer’s comment . In the manuscript, however, the discussion is inserted in the Results and discussion.

In this regard, we have added some sentences and new references as follows:

Indeed, he used foliar fertilizers, containing  macromolecular compounds, in addition to the typical chemical elements (carbon, oxygen, hydrogen, nitrogen, sulfur), influenced positively the tree growth as they enhance metabolism and activity of plant photosynthesis L 305 (Mihalache, 2017)

Generally, the best results obtained from the application of above biostimulants in 2019, although their nature is diverse, are to be ascribed to the more stressful climatic conditions that occurred in this season, confirming their action for mitigating the adverse impacts of harsh environmental conditions on plants (Bhupenchandra). L 214  

Mihalache D., Sirbu C., Grigore A ,Stănescu A.M., Calciu I.C. , Marin N., 2017. Physical, chemical and agrochemical characterization of some organo-mineral fertilizers Romanian Biotechnological Letters, 22, 1, 12259-12666.

Point 2. Furthermore, the authors must interpret the effect of seasons on the characteristics under

consideration.

We agree with the reviewer’s comment

We have added the following sentences

“The difference in climatic conditions over the years has influenced the various production parameters and fruit quality. In particular, in the second year of the trial, characterized by higher maximum temperatures, greater windiness and drier conditions, the size of the fruit and the individual constituents of the fruit itself tended to be smaller, on the contrary, however, the physical-chemical parameters were more higher than in 2018. These results have also been confirmed previously by other studies on pomegranate when extreme hot and dry climate during the period of maximal growth rate was associated with restricted growth and a high proportion of small-size fruit. (Joshi et al.,2021

Joshi M., Ze’ev Schmilovitch Z. and  Ginzberg I.  Pomegranate Fruit Growth and Skin Characteristics in Hot and Dry Climate Front. Plant Sci., 19 August 2021 https://doi.org/10.3389/fpls.2021.725479

Best regards

Annalisa Tarantino

Reviewer 2 Report

The manuscript by authors Annalisa Tarantino et al. titled „Foliar application effect of organo mineral fertilizers with biostimulating action on the vegetative growth, yield, and quality of wonderful pomegranate fruits in two consecutive years“ deals with a relevant and novel topic. The introduction sufficiently describes the current state of the problem and the rationale of the research. The material and methods are described with reserves. The study contains adequate results that are appropriately discussed. The conclusion reflects the most important results achieved. I have some well-intentioned advice on the manuscript that could improve its quality and suitability for publication:

1.     The manuscript lacks major chapter headings - for example, Material and Methods.

2.     For this type of field experiment, I recommend adding more detailed soil characteristics (soil sample analyses) and detailed weather conditions course (temperature and precipitation) to the manuscript. The above will allow a better understanding of some of the relationships in the results of the study. In this context, I recommend that Chapter 3.1 should be moved to the methodology chapter.

3.     All material used should be specified in more detail (commercial name, manufacturer, country of origin) and all methods used should be supported by citations, including cultivation technology and experimental design.

4.     For Figure 2, I would kindly ask the authors to consider using thinner curves in the graph for better readability.

Author Response

Revision of the manuscript “Foliar application effect of organo mineral fertilizers with biostimulating action on the vegetative growth, yield, and quality of wonderful pomegranate fruits in two consecutive years”, by Annalisa Tarantino, Grazia Disciglio , Laura Frabboni and Giuseppe Lopriore.

Dear Editor,

We appreciate the opportunity to revise the above-mentioned manuscript submitted to Horticulturae. We sincerely thank you and the Reviewers for the effort and constructive remarks.

Below we have addressed the reviewers’ comments. Additions to the manuscript have been highlighted in green in this letter, while in red in the revised manuscript.

Sincerely yours,

Dr. Annalisa Tarantino

e-mail: annalisa.tarantino@unifg.it

reviewer 2

Comments to Author

1The manuscript lacks major chapter headings - for example, Material and Methods.

 The authors  apologize to the reviewer for the mistake.

We have inserted the term jn the chapterMaterials and methods”

  1. For this type of field experiment, I recommend adding more detailed soil characteristics (soil sample analyses) and detailed weather conditions course (temperature and precipitation) to the manuscript. The above will allow a better understanding of some of the relationships in the results of the study. In this context, I recommend that Chapter 3.1 should be moved to the methodology chapter.

We thank the Reviewer for these considerations

We have inserted  in the manuscript the table 1 of climatic parameters and the soil physic-chemical parameters as follows:

                                                   Table 1 - Monthly mean maximum and minimum temperatures (Tmax, Tmin) and

relative air humidity (RHma  and RHmin), wind speed (Ws), radiation (Rad) and total precipitation (P) in 2018 and 2019 seasons.

Month

Tmax

Tmin

RHmax

RHmin

Ws

Rad

P

(°C)

(°C)

(%)

(%)

(m sec-1)

(Wm-2)

(mm)

2018

April

21.3

12.9

94.6

37.6

2.8

235.3

54.0

May

26.1

13.4

95.2

49.1

2.4

275.8

58.3

June

30.0

12.1

89.5

40.3

3.4

289.6

88.2

July

33.3

19.6

83.6

35.4

3.0

318.7

16.8

Aug

32.7

20.1

71.3

28.3

2.1

285.7

39.1

Sept

29.1

17.1

81.3

30.0

3.7

193.6

80.0

Mean

23.7

15.9

85.9

36.8

2.9

266.5

Total

366.4

2019

April

20.6

8.2

94.4

51.0

3.7

190.2

40.3

May

21.3

10.2

95.3

56.3

4.0

232.9

86.7

June

33.2

17.5

85.9

35.1

3.7

252.2

9.2

July

33.7

19.5

84.0

33.9

3.7

258.8

30.0

Aug

34.8

20.3

79.9

33.9

3.6

225.6

5.7

Sept

29.5

16.8

88.7

42.6

3.6

175.5

3.8

Mean

28.8

15.4

88.0

42.5

3.7

222.5

Total

175.7

 The soil is a silty-clay vertisol of alluvial origin (1.20 m depth) (Typic Chromoxerert, fine, thermic, according to the Soil Taxonomy-USDA), with the following characteristics: 32.7% clay; 30.5% silt; 36.8 % sand; pH 8.8 (soil:water 1:2.5); 1.4% organic matter; 1.5‰ total N, 9.7% active CaCO3. 56 mg kg–1 P205 and 1390 mg kg–1 exchageable K.

In the Results we have also added the following sentences and new references:

 “The difference in climatic conditions over the years has influenced the various production parameters and fruit quality. In particular, in the second year of the trial, characterized by higher maximum temperatures, greater windiness and drier conditions, the size of the fruit and the individual constituents of the fruit itself tended to be smaller, on the contrary, however, the physical-chemical parameters were more higher than in 2018. These results have also been confirmed previously by other studies on pomegranate when extreme hot and dry climate during the period of maximal growth rate was associated with restricted growth and a high proportion of small-size fruit. (Joshi et al.,2021)

The Chapter 3.1 has been moved to the methodology chapter.

3 . All material used should be specified in more detail (commercial name, manufacturer, country of origin) and all methods used should be supported by citations, including cultivation technology and experimental design.

Thank you for your observation, we have  added ln the manuscript the following sentences:

The commercial name and  manufacturer of  products used  are specified in tab.1 of the manuscrips,as follows;

 ERGOSTIM XL® (Isagro), SIAPTON 10L® (Siapa), ALLIBIO-RAD ® (Fertek) :

The experiment set up was organized as a completely randomized block design with four replications per treatment and five trees per replicate. Trees were planted at a distance of 3 m on the row and 5.5 m between the rows. To avoid any contamination between treatment one buffer row was located
between replicates and blocks, along with two or more buffer rows around the perimeter of the experimental field.  In each replicate three centrally located plants were used to collect vegetative and reproductive parameters

We have  added in the manuscript also the following sentence

Pomegranate trees were  drip irrigated and received the ordinary cultural practices (pruning, fruit thinning, fertilization, pest, and disease protection) of the area. In particular, for fertilization schedules, the doses per year applied in the experimental field were field were 115, 90 and 120 kg ha−1, respectively, for N, P 2O 5 and K2 O. The dose of nitrogen was divided into 3 from April to August. Phosphorus and potassium were administered annually in winter.

  1. For Figure 2, I would kindly ask the authors to consider using thinner curves in the graph for better readability.

Thank you for your observation

“We have tried to thin curves  but they didn’t improve”.

Best regards

Annalisa Tarantino

Round 2

Reviewer 1 Report

The authors have been addressed all the comments

Author Response

Dear reviewer,

Thanking you very much, once again, for your contribution to the review of the article, in which, after accepting all your suggestions, point by point, I sincerely apologize for the errors that occurred in sending the article without including the suggestions and corrections as well as those of MDPI reviewers in English. Therefore, in the attached article I have included all the corrections and additions recommended by you. Furthermore, I have inserted a new title proposed by the editor, and eliminated the repetitions previously present in the text.

 Best regards,

Annalisa Tarantino

Reviewer 2 Report

Based on a careful study of Annalisa Tarantino et al.'s responses to my first review, I can say that the authors incorporated all of my well-meaning comments into the manuscript. I now recommend the manuscript for publication. I wish the authors all the best in further interesting research.

Author Response

Thank you once again for your contribution as a reviewer

Best regards

Annalisa Tarantino